# Conformational dynamics of auto-inhibition in the ER calcium sensor STIM1

Stijn van Dorp[1], Ruoyi Qiu[1], Ucheor B Choi[1†], Minnie M Wu[1], Michelle Yen[1], Michael Kirmiz[1], Axel T Brunger[1,2], Richard S Lewis[1]*

[1]Department of Molecular and Cellular Physiology, Stanford University School of Medicine, Stanford, United States; [2]Howard Hughes Medical Institute, Stanford University School of Medicine, Stanford, United States

**Abstract** The dimeric ER Ca$^{2+}$ sensor STIM1 controls store-operated Ca$^{2+}$ entry (SOCE) through the regulated binding of its CRAC activation domain (CAD) to Orai channels in the plasma membrane. In resting cells, the STIM1 CC1 domain interacts with CAD to suppress SOCE, but the structural basis of this interaction is unclear. Using single-molecule Förster resonance energy transfer (smFRET) and protein crosslinking approaches, we show that CC1 interacts dynamically with CAD in a domain-swapped configuration with an orientation predicted to sequester its Orai-binding region adjacent to the ER membrane. Following ER Ca$^{2+}$ depletion and release from CAD, cysteine cross-linking indicates that the two CC1 domains become closely paired along their entire length in the active Orai-bound state. These findings provide a structural basis for the dual roles of CC1: sequestering CAD to suppress SOCE in resting cells and propelling it toward the plasma membrane to activate Orai and SOCE after store depletion.

*For correspondence:
rslewis@stanford.edu

Present address: †Department of Biochemistry, West Virginia University School of Medicine, Morgantown, United States

## Editor's evaluation

This study uses complementary approaches to advance our mechanistic understanding of STIM1 activation, with elegant single molecule methods providing new details on STIM1 structure and dynamics. The data clarifies some of the controversy between domain packing in two differing X-ray and NMR structures and substantially contributes to a mechanistic and structural understanding of the STIM1 activation process.

## Introduction

Store-operated Ca$^{2+}$ entry (SOCE) is a nearly ubiquitous signaling pathway activated by extracellular stimuli that deplete Ca$^{2+}$ from the endoplasmic reticulum (ER). SOCE is essential for diverse physiological functions of excitable and non-excitable cells, including gene expression, secretion, and motility (*Prakriya and Lewis, 2015*). Accordingly, loss-of-function mutations in the core SOCE components lead to serious human pathologies such as severe combined immunodeficiency, autoimmunity, myopathy, ectodermal dysplasia, and anhidrosis (*Lacruz and Feske, 2015*), whereas gain-of-function mutations cause Stormorken syndrome, miosis, myopathy, thrombocytopenia, and excessive bleeding (*Böhm and Laporte, 2018*). These clinical manifestations underscore the need for precise regulation of SOCE to ensure that it is silent when ER Ca$^{2+}$ stores are full yet reliably activated by stimuli that drive store depletion.

The dimeric ER membrane protein STIM1 (*Figure 1A*, top) controls SOCE through the regulated interaction of its CRAC activation domain (CAD [*Park et al., 2009*], also known as SOAR [*Yuan et al., 2009*] or CCb9 [*Kawasaki et al., 2009*]) with Orai1 Ca$^{2+}$ channels in the plasma membrane. In resting cells, Ca$^{2+}$ bound to the luminal EF hands of the STIM1 dimer suppresses STIM1 activity by promoting

**Figure 1.** Parallel orientation of CC2 domains in ctSTIM1 is consistent with the CAD crystal structure. (**A**) (*top*) Schematic overview of domains in ctSTIM1, comprising the membrane-proximal coiled-coil 1 (CC1) domain with helical regions α1 (aa 238–271), α2 (aa 278–304), and α3 (aa 308–337), and the CC2 (345–391) and CC3 (aa 408–437) domains which are part of CAD. (*bottom*) The orientations of the CC2 domains distinguish the NMR structure of CC1α3-CC2 (*bottom left*, anti-parallel CC2 domains; 2MAJ.pdb) from the crystal structure of CAD (*bottom right*, parallel CC2 domains; 3TEQ.pdb). Sites tested by inter-subunit smFRET are indicated. Although aa 337 is not part of the crystal structure (aa 344–443), it is sufficiently close to permit comparison to the NMR structure. (**B**) Two methods for immobilizing single ctSTIM1 dimers for smFRET. ctSTIM1 peptides were encapsulated in liposomes and immobilized on a glass slide by a biotin-neutravidin interaction or were attached directly via a C-terminal avitag. (**C**) Example recordings of donor (*green*) and acceptor (*red*) fluorescence evoked by donor excitation for sites 337:337′, 363:363′, and 378:378′. Single-step bleaching events indicate a single molecule, while the anti-correlated response of the donor to the acceptor bleach event is a hallmark of smFRET. Calculated FRET ratios (*black*) are shown below. (**D**) Ensemble smFRET histograms for sites 337:337′, 363:363′, and 378:378′ displayed predominantly high FRET, indicating close parallel apposition. (**E**) A comparison of smFRET-derived distances (*black*) with simulated inter-fluorophore distances from the crystal (*white*) and NMR structures (*gray*). The measured distances closely match the crystal structure but deviate strongly from the NMR structure, particularly for 337:337′ and 378:378′. smFRET error bars indicate the expected distance deviation corresponding to an uncertainty of ±0.05 in FRET measurement. 'Crystal' and 'NMR' error bars indicate the s.e.m. of inter-fluorophore distance from 100 simulations of dye position (see Materials and methods).

The online version of this article includes the following figure supplement(s) for figure 1:

**Figure supplement 1.** smFRET histograms for CAD measurements.

**Figure supplement 1—source data 1.** Time series of donor and acceptor fluorescence and FRET for single molecules.

**Figure supplement 2.** smFRET histograms for CC1 measurements.

**Figure supplement 2—source data 1.** Time series of donor and acceptor fluorescence and FRET for single molecules.

**Figure supplement 3.** smFRET histograms for CC1:CAD measurements.

**Figure supplement 3—source data 1.** Time series of donor and acceptor fluorescence and FRET for single molecules.

intramolecular association of the coiled-coil 1 (CC1) domains with CAD, referred to as the 'inhibitory clamp' (*Korzeniowski et al., 2010*; *Yang et al., 2012*; *Yu et al., 2013*; *Fahrner et al., 2014*; *Ma et al., 2015*). Upon ER Ca²⁺ depletion, dissociation of Ca²⁺ from the luminal EF hands triggers conformational changes that release CC1 from CAD and promote STIM1 accumulation at ER-plasma membrane (ER-PM) junctions where the CAD domain binds, traps, and activates Orai1 channels diffusing in the PM (*Wu et al., 2006*; *Wu et al., 2014*; *Liou et al., 2007*; *Park et al., 2009*; *Yuan et al., 2009*). Thus, to elucidate the regulatory mechanism of SOCE, it is essential to understand how CC1 interacts with CAD to control STIM1 activity.

The CC1 domain comprises three helical segments, CC1α1–3 (*Soboloff et al., 2012*; *Figure 1A*). Current evidence supports a model in which CC1α1, the most ER-proximal segment, acts as a bimodal switch to stabilize both the quiescent and active states of STIM1. According to this model, in cells with full ER Ca²⁺ stores, CC1α1 binds to the CAD CC3 helix to sequester CAD (*Fahrner et al., 2014*; *Ma et al., 2015*); upon store depletion, the luminal EF-SAM domains dimerize, CC1α1 unbinds from CAD, and the transmembrane and proximal CC1α1 regions form a coiled-coil (*Stathopulos et al., 2006*; *Hirve et al., 2018*; *Schober et al., 2019*), projecting CAD toward the plasma membrane. Although several residues in CC1α1 and CC3 have been identified as essential for forming the inhibitory clamp (*Muik et al., 2011*; *Zhou et al., 2013*; *Fahrner et al., 2014*; *Ma et al., 2015*), the helical arrangement of the CC1α1:CAD binding interface is completely unknown. The CC1α3 domain is also implicated in stabilizing the inactive state (*Korzeniowski et al., 2010*; *Yang et al., 2012*), but it is unclear whether this involves binding to CAD or another mechanism (*Zhou et al., 2013*).

Attempts to determine the structure of full-length STIM1 in the inactive or active state have been unsuccessful. Instead, structures of several fragments from the STIM1 cytosolic region have offered potential clues, although it is not yet clear how they might relate to physiological states of the full-length protein. The crystal structure of human STIM1 CAD (aa 345–444, with mutations L374M, V419A, and C437T) depicts a parallel V-shaped dimer comprising the CC2 and CC3 domains (*Figure 1A*, right; *Yang et al., 2012*). It is unknown whether this structure represents an active or inactive conformation, and the absence of CC1 precludes inferences about its interaction with CAD. A solution NMR structure of a CC1α3-CC2 fragment depicts a dimer of wedged antiparallel CC2 helices (*Figure 1A*, bottom left; *Stathopulos et al., 2013*). This NMR structure can bind Orai1 C-terminal fragments in vitro and has been proposed to be an intermediate in the activation of STIM1 (*Stathopulos et al., 2013*), but it has never been shown to bind intact Orai1 channels in situ and lacks the CC3 helix needed to evoke SOCE (*Covington et al., 2010*). Finally, two contrasting structures have been reported for the conformation of the CC1 region. In a crystal structure of CC1 (aa 237–340, with mutations M244L and L321M), two elongated CC1 helices form an antiparallel coiled-coil dimer within the CC1α2 and CC1α3 domains (*Cui et al., 2013*; see Figure 5C ), but it is unclear how this elongated structure with its widely separated C-termini could attach to the closely paired N-termini of CAD. In contrast, a recent solution NMR study of CC1 (aa 234–343) depicts a compactly folded monomeric structure in which CC1α1-α3 form a tight three-helix bundle (*Rathner et al., 2021*).

Although the sequences of these various structures partially overlap, their conformations are incompatible with each other, raising fundamental questions about their potential relation to the inactive and active forms of STIM1. What is the conformation of CAD in the context of the entire cytosolic domain? How is CC1 arranged relative to CAD in the inactive state, and how does this change during activation? Which regions of STIM1 are stable, and which are flexible? To address these questions, we examined the structure and dynamics of the full cytosolic domain of STIM1 (ctSTIM1, aa 233–685) using single-molecule Förster resonance energy transfer (smFRET) measurements at sites throughout CC1 and CAD. smFRET is commonly applied to estimate intramolecular distances and reveal asynchronous molecular transitions that are typically obscured in population measurements. Our results show that CAD in ctSTIM1 resembles the CAD crystal structure, but with unexpected structure and flexibility in an apical region that is critical for Orai binding and activation. CC1α1 interacts with CAD in a domain-swapped configuration to form the inhibitory clamp, while the Stormorken syndrome mutation R304W releases CC1α1 from CAD to activate ctSTIM1. In studies of full-length STIM1 in intact cells, cysteine crosslinking after store depletion suggests that after unbinding from CAD, the two CC1 domains pack together along their entire length, in contrast to the CC1 crystal and CC1α3-CC2 NMR structures. Together, these findings offer the first structural view of the STIM1 inhibitory clamp and

reveal the massive conformational changes evoked by store depletion, which serve to reorient and translocate CAD towards the PM where it can bind Orai1 and activate SOCE.

## Results

### The CC2 domains in ctSTIM1 are oriented in a parallel configuration

We initially applied smFRET to test whether the conformation of CAD within the cytosolic domain (ctSTIM1, aa 233–685) resembles that of the CC2-CC3 crystal structure (*Yang et al., 2012*) or the CC1α3-CC2 NMR structure (*Stathopulos et al., 2013*), referred to below as the 'crystal structure' or 'NMR structure' respectively. A major distinction between the two structures is that the CC2 domains (aa 345–391) in CAD are parallel in the crystal structure but antiparallel in the NMR structure (*Figure 1A*). After removing the single native cysteine in ctSTIM1 with a C437S mutation, we created three mutants with cysteines at positions 337, 363, and 378, for which the two structures predicted very different inter-subunit distances, particularly for 337:337′ and 378:378′ (hereafter ':' is used to denote a pair of sites in the dimer and '′' indicates a site on the adjacent subunit) (*Figure 1A*). The ctSTIM1 mutants were expressed in *E. coli* and purified, yielding full-length protein of >90% purity (see Materials and methods). After randomly labeling the cysteine-mutant ctSTIM1 molecules with FRET donor (Alexa Fluor 555) and acceptor (Alexa Fluor 647) dyes, single ctSTIM1 dimers were surface-tethered to a glass coverslip for imaging by TIRF microscopy, either encapsulated in liposomes or attached directly through a tag on the C-terminus (*Figure 1B*). In several instances we tested, both attachment methods yielded similar results (*Figure 1—figure supplements 1–3*); liposome encapsulation was used for all measurements except intramolecular FRET measurements in *Figure 2* and 4 (see Materials and methods, 'Sample preparation' for more details). Upon excitation of the donor dye, the donor and acceptor emissions of individual molecules were ratioed to calculate FRET efficiencies, which were high and stable at all three label sites (*Figure 1C and D*). These results are consistent with a CAD conformation in which the CC2 domains are closely apposed and parallel, as in the crystal structure. Inter-fluorophore distances calculated from simulations of the donor and acceptor dye positions on the crystal structure closely matched those derived from smFRET, while simulations based on the NMR structure deviated substantially (*Figure 1E*). We conclude that in the context of the complete ctSTIM1 in solution, the CC2 domains in CAD adopt a parallel conformation consistent with the CAD crystal structure and not the NMR structure.

### The apex of CAD deviates from the crystal structure

We made additional smFRET measurements throughout the entire CAD to allow a more complete comparison with the crystal structure. Near the C-terminal end of the CC3 domain (aa 408–437), the 431:431′ label pair produced a stable, narrowly defined high-FRET state (*Figure 2A*), and inter-subunit smFRET throughout the CAD exceeded 0.5, consistent with a compact parallel dimer (*Figure 2B*). Near the dimer interface (hereafter termed the 'base' of CAD), smFRET-derived distances quantitatively agreed with those from the crystal structure, and an intra-subunit measurement 431:363 directly confirmed the compact folding of CC3 onto CC2 within each subunit (*Figure 2C*).

In contrast to the base of CAD, smFRET measurements of the distal CC2 and CC2-CC3 linker regions (the CAD 'apex') deviated from those indicated in the crystal structure (*Figure 2D*). This is particularly interesting as mutagenesis studies have identified this region as critical for Orai1 binding and activation (*Calloway et al., 2010*; *Korzeniowski et al., 2010*; *Wang et al., 2014*; *Thompson et al., 2018*; *Butorac et al., 2019*). To provide a rough indication of the extent of the deviation implied by smFRET, we generated an alternative model of the CAD apex based on the crystal structure by allowing rotation around G379 to bring the simulated FRET efficiencies in line with our experimental results. In the resulting model the distal CC2 regions were rotated outward, while the regions around aa 400 were pulled closer together, effectively compacting the apical 'V-shape' (*Figure 2E* and see Materials and methods). Whereas inter-subunit smFRET at aa 399–401 appeared as a well-defined peak, FRET levels at the distal end of CC2 were broadly distributed and appeared multimodal, particularly at aa 388 (*Figure 2B*). Accordingly, smFRET measurements in this region displayed fluctuations (*Figure 2F*), suggesting conformational flexibility that was not expected from the uninterrupted rigid helix of the crystal structure. These differences presumably arose from stabilization of an alternate apical conformation in the crystal by hydrogen bonds between the apex and adjacent CAD subunits

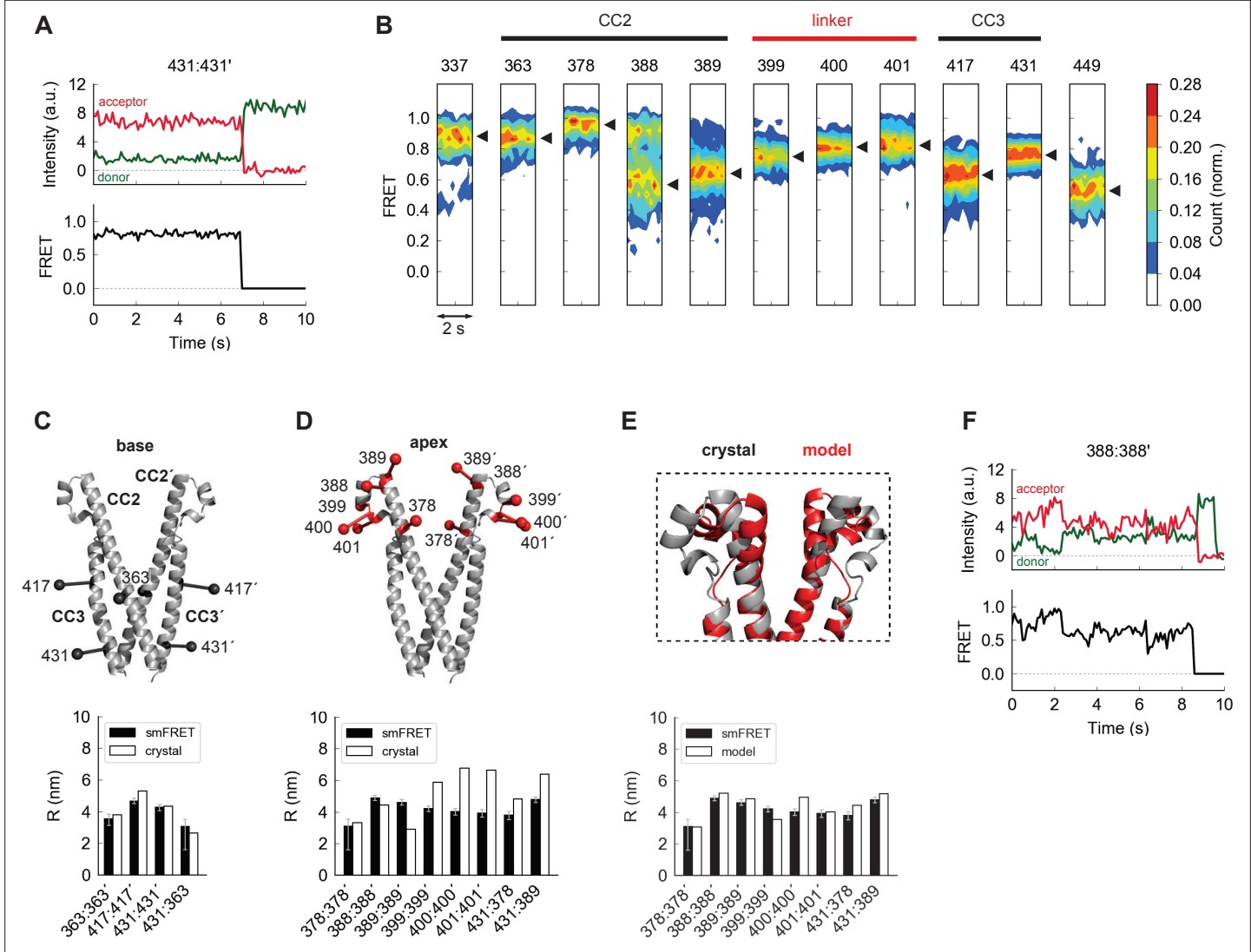

**Figure 2.** The apex of CAD in ctSTIM1 deviates from the CAD crystal structure. (**A**) Representative smFRET recording at the base of CAD (431:431′) showing stable, high FRET. (**B**) Ensemble density plots of the initial 2 s of inter-subunit smFRET recordings for sites throughout CAD. Predominant FRET levels are shown by arrowheads to the right of each plot. (**C–D**) Comparison of inter-fluorophore distances from smFRET and the CAD crystal structure. Simulated fluorophores are represented by the average position of a central atom, indicated here as ball-on-stick models (see Materials and methods). For sites near the dimer interface (the 'base' of CAD, **C**), the smFRET-derived distances (*black bars*) agree closely with the crystal structure (*white bars*). Measurements in the apex (**D**) deviate from the crystal structure. (**E**) Modified structure (*red*) generated by a smFRET-constrained optimization of the crystal structure (*gray*). Outward rotation of the distal CC2 region (aa 379–391) around G379 as a pivot point restored close correspondence between smFRET-derived distances and the crystal structure (*bottom*). (**F**) Representative example of smFRET fluctuations in the CAD apex (388:388′).

The online version of this article includes the following figure supplement(s) for figure 2:

**Figure supplement 1.** CAD apex structure.

**Figure supplement 1—source data 1.** Raw unedited gel for *Figure 2—figure supplement 1A* (ctSTIM1 CAD cysteine crosslinking).

**Figure supplement 1—source data 2.** Uncropped labeled gel for *Figure 2—figure supplement 1A* (ctSTIM1 CAD cysteine crosslinking).

(*Figure 2—figure supplement 1B*). The smFRET-derived model in *Figure 2E* thus likely represents just one of a number of potential apical conformations sampled by spontaneous fluctuations of the distal CC2 domain.

As a complementary structural test, we analyzed inter-subunit cysteine-cysteine (cys-cys) cross-linking upon oxidation by copper phenanthroline (CuP, see Materials and methods). Cysteines introduced throughout the CAD apex formed inter-subunit crosslinks, detected as ctSTIM1 dimers on non-reducing SDS-PAGE gels (*Figure 2—figure supplement 1A*). This result shows that opposing

residues in the apical region at least transiently come within the ~6 Å range for cys-cys crosslinking (*Qin et al., 2015*), again inconsistent with the CAD crystal structure. Moreover, crosslinking efficiencies were similar for three adjacent apical cysteine pairs (399:399′, 400:400′, and 401:401′), suggesting significant rotational flexibility beyond the 100 ms temporal resolution of our smFRET recordings. Taken together, these data show that flexibility and spontaneous fluctuations cause the CAD apical region to deviate significantly from the crystal structure, unlike the stable basal region.

## CC1α1 docks parallel to CC3 in a domain-swapped configuration

The CC1 domain controls the activation of STIM1 in vivo by releasing CAD for interaction with Orai1 (*Zhou et al., 2013*; *Ma et al., 2015*). Although it is clear that the membrane-proximal CC1α1 domain can interact with CAD to inhibit STIM1 (*Ma et al., 2015*), their relative orientations remain undefined. We detected high inter-subunit FRET efficiencies between labels at the N-termini of CC1α1 and CC3 (red histogram in *Figure 3A*, top), as well as between labels at their C-termini (blue histogram in *Figure 3A*, bottom), indicating that the CC1α1 domain (a continuous α-helix [*Cui et al., 2013*; *Rathner et al., 2021*]) is oriented parallel to CC3 in the predominant ctSTIM1 conformation.

To determine whether CC1α1 associates with the CAD of the same subunit or the adjacent one within the dimer, we compared inter-subunit and intra-subunit FRET efficiencies between labels at the N-terminus of CC1α1 and three sites in CAD (*Figure 3B*). The predominant inter-subunit FRET efficiencies were consistently higher than intra-subunit FRET efficiencies, demonstrating that in the predominant ctSTIM1 conformation, CC1α1 engages CAD in a domain-swapped configuration. We also observed spontaneous and reversible transitions from the predominant FRET state, in which FRET efficiencies fluctuated between the same two levels for both inter- and intra-subunit measurements (*Figure 3C*). Similar fluctuation behavior and transition kinetics were observed at all three CAD sites (*Figure 3—figure supplement 1*), suggesting a conformational transition in which the two CC1α1 domains briefly switch sides on CAD, probably using the same binding interface (*Figure 3D*).

## The CC1α3 domains are compactly folded and directed away from CAD

In vivo, mutations or deletions of the CC1α3 domain activate STIM1 (*Korzeniowski et al., 2010*; *Yang et al., 2012*) but the underlying mechanism, in particular the proposal that CC1α3 interacts directly with CAD (*Yang et al., 2012*), has been questioned (*Zhou et al., 2013*). We measured predominantly high inter-subunit FRET efficiencies of ~0.9 between labels at the N- or C-terminal ends of the CC1α3 domains (aa 312 and 337, respectively), indicating that the two domains are closely apposed and parallel (*Figure 4A*). To determine their orientation relative to CAD, we measured intra-subunit FRET efficiencies between labels at either end of the CC1α3 domain to three sites in CC2. While FRET efficiency from the CC1α3 C-terminus to the proximal CC2 domain was high as expected (red histogram in *Figure 4B*, top), low FRET efficiencies were measured from the CC1α3 N-terminus, indicating that the N-terminus is directed away from CAD (*Figure 4B*, bottom).

Stable high inter-subunit FRET efficiencies at the CC1α3 C-terminus (aa 337) were consistent with its proximity to the compact, stable CAD base. However, label sites near the N-terminus of CC1α3 displayed brief (0.8 s average lifetime), large transitions to a low FRET state, suggesting that the N-termini of CC1α3 intermittently splay out from a stable pivot point at the base of CAD (*Figure 4C and D*). While these large conformational transitions were most prominent, further analysis revealed additional substates in the smFRET trajectories (*Figure 4—figure supplement 1*).

## A model for the CC1-CAD complex

Through the integration of many inter-molecular distance measurements, smFRET can be applied to build models of tertiary protein structure (*Brunger et al., 2011*). We used 36 unique inter- and intra-subunit smFRET measurements (*Supplementary file 1*) to develop a structural model of the resting conformation of CC1 subdomains relative to the optimized model of CAD depicted in *Figure 2E*. CC1 in the ctSTIM1 dimer was assumed to comprise three rigid α-helical sections (CC1α1-α3, comprising residues aa 246–271, 275–305, and 310–337, respectively) connected by flexible linkers, consistent with bioinformatic analysis (*Soboloff et al., 2012*), NMR measurements of the CC1 fragment (*Rathner et al., 2021*), and distances estimated from intra-subunit smFRET efficiencies (239:274, 274:307, and 307:337; *Figure 1—figure supplement 2*). We selected randomly generated models that met the criteria of satisfying smFRET-derived distance constraints while avoiding steric clashes with each other

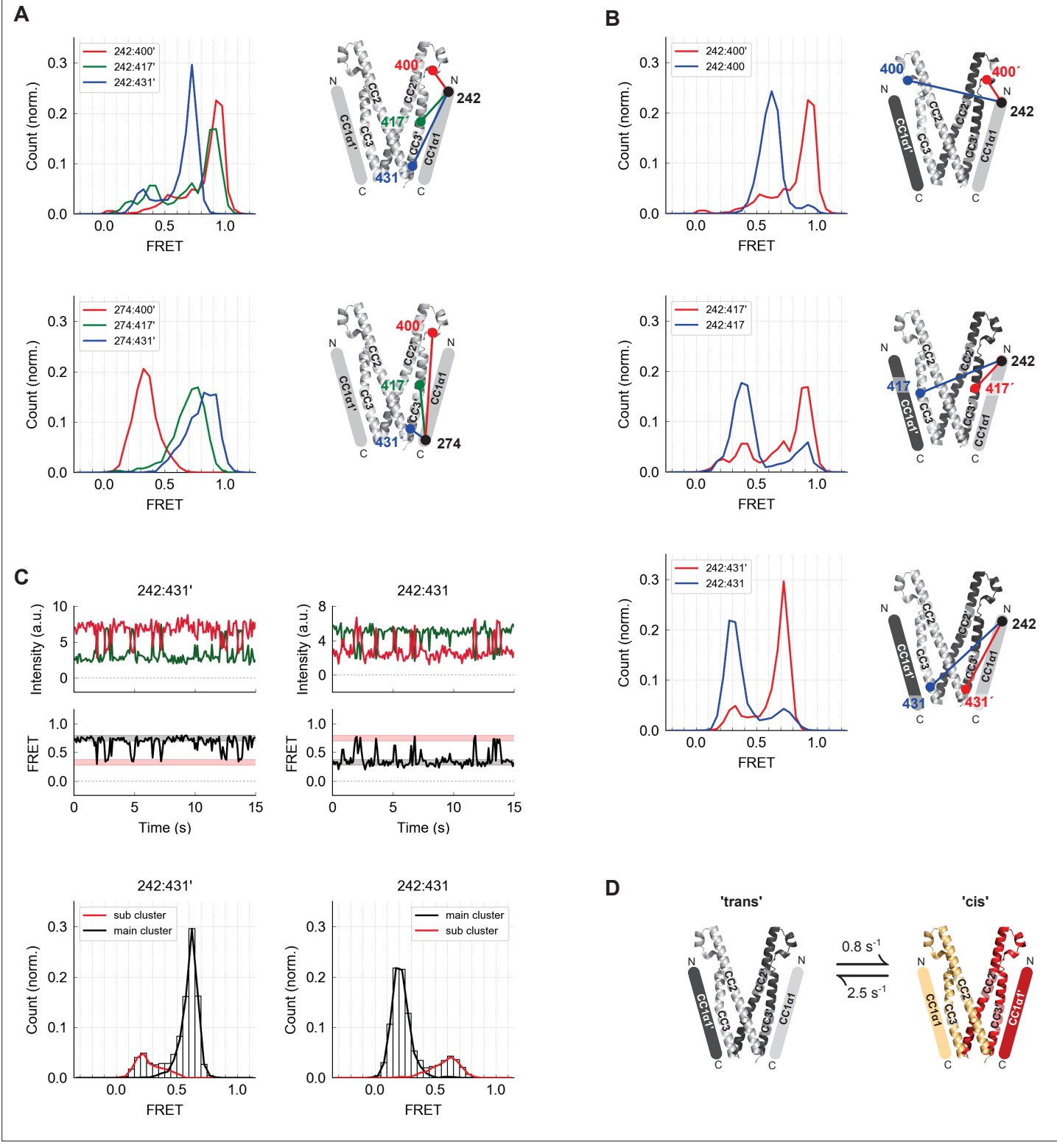

**Figure 3.** The CC1α1 domain is near and parallel to the CC3 domain of the opposing CAD subunit. (**A**) Asymmetric inter-subunit smFRET measurements between CC1α1 and CC3 reveal a parallel orientation, with high FRET levels at 242:400´ (*top, red*) and 274:431´ (*bottom, blue*) indicating close apposition. The crystal structure of CAD is shown with schematic representation of CC1α1 domains. (**B**) The predominant smFRET level between CC1α1 and CC3 was consistently higher for inter-subunit measurements (*red*) than intra-subunit measurements (*blue*). This pattern indicates a predominant domain-swapped configuration in which CC1α1 docks close to the CC3 domain of its opposing subunit. (**C**) Inter- and intra-subunit recordings (242:431´ and 242:431, respectively) displayed brief smFRET transitions between two states indicated by red and black bars. In

*Figure 3 continued on next page*

*Figure 3 continued*

both cases, fluctuations occurred between the same two FRET levels, reflected in the main and sub cluster peaks of the ensemble FRET histogram below, suggesting the CC1α1 domains switched sides on CAD (see also *Figure 3—figure supplement 1*). (**D**) Schematic illustration of spontaneous conformational transitions of the CC1α1 domain. In the predominant conformation, CC1α1 is closely apposed and parallel to CC3 on its opposing subunit in a domain-swapped configuration ('trans'). Occasionally, CC1α1 switches to interact briefly with CC3 on its own subunit ('cis'). Rate constants were derived from dwell time analysis of smFRET traces (*Figure 3—figure supplement 1E,F*).

The online version of this article includes the following figure supplement(s) for figure 3:

**Figure supplement 1.** CC1α1:CAD dynamics.

**Figure supplement 1—source data 1.** Time series of FRET, Hidden-Markov Model fits, and state clusters for single molecules.

or with CAD (see Materials and methods and *Figure 5—figure supplement 1A* for details of the modeling process). Two classes of models emerged that were highly similar but distinguishable by the organization of the CC1α2/α3 domains. In ~70 % of solutions, the CC1α2/α3 domains were stacked parallel (average model in *Figure 5A* from an ensemble of models in *Figure 5—figure supplement 1B*), while the remaining solutions had an alternative, wedged CC1α2/α3 topology (*Figure 5A* and *Figure 5—figure supplement 1C*). Importantly, both classes of solutions displayed the defining features described above, including closely apposed CC1α3 domains directed away from CAD and CC1α1 domains oriented parallel to CC3 in a domain-swapped configuration. As an independent test of the smFRET-derived models, we applied the lysine crosslinking agent BS[3] combined with mass spectrometry (XLMS) to identify proximal pairs of residues (*Rappsilber, 2011*). BS[3] XLMS confirmed key structural arrangements of both models of the CC1-CAD complex, including the compact packing of CC1α2/α3 domains and the domain-swapped CC1α1-CAD interaction (*Figure 5—figure supplement 2*).

Although the models do not specify the orientation of side chains in the CC1 helices, they highlight two regions for potential inter-subunit coiled-coil interactions in the CC1-CAD complex. The antiparallel apposition of hydrophobic residues in CC1α2 and CC1α3′ (*Figure 5B*) is reminiscent of the tight, antiparallel interaction of CC1α2 and CC1α3′ domains in the CC1 dimer crystal structure (*Cui et al., 2013*), an arrangement that was further supported by the formation of amine-carboxy crosslinks in ctSTIM1 by 1-ethyl-3-(3-dimethylaminopropyl)carbodiimide hydrochloride (EDC), a zero-length crosslinker (*Figure 5C* and *Figure 5—figure supplement 3*). Interestingly, the CC1α1:CC3′ interface in the smFRET-derived models juxtaposes multiple hydrophobic residues that have been previously implicated through mutagenesis in stabilizing the CC1-CAD complex: L248, L251, L258, and L261 in CC1α1 and L416, V419, and L423 in CC3 (*Muik et al., 2011*; *Zhou et al., 2013*; *Fahrner et al., 2014*; *Ma et al., 2015*; *Figure 5D*). To refine the CC1α1:CC3′ interface, we aligned the crystal structure of the CC1α1 domain (aa 246–271) to the smFRET-derived model and performed a computational docking simulation using Rosetta software (see Materials and methods). The resulting model (*Figure 5E*) indicates an extensive pairing of hydrophobic residues predicted to stabilize the CC1α1:CC3′ interface, providing a plausible explanation for the activating effects of disruptive mutations at these sites.

## The Stormorken mutation R304W activates ctSTIM1 by releasing CC1α1 from CAD

The close packing of CC1α1 and CAD in our model suggests that ctSTIM1 is primarily inactive. Accordingly, when expressed with Orai1 in HEK293 cells, mCh-ctSTIM1 only slightly increased [Ca$^{2+}$]$_i$ compared to mCh-CAD (*Figure 6—figure supplement 1*), in agreement with previous evidence that ctSTIM1 is a relatively weak activator of Orai1 (*Muik et al., 2009*; *Park et al., 2009*; *Yuan et al., 2009*; *Korzeniowski et al., 2010*).

Deletion of the CC1 domain (mCh-ctSTIM1-ΔCC1) restored ctSTIM1 activity to the level seen with CAD, as did the introduction of R304W, a naturally occurring mutation that causes Stormorken syndrome by constitutively activating STIM1 (*Misceo et al., 2014*; *Morin et al., 2014*; *Nesin et al., 2014*; *Figure 6—figure supplement 1*). Interestingly, introduction of R304W into ctSTIM1 almost completely eliminated the high inter-subunit FRET between the N termini of CC1α1 and CC3 in vitro, indicating the release of CC1α1 from CAD (*Figure 6A*, aa 242:400′). A more moderate decrease in FRET efficiency was observed between CC1α1 and the CC3 C-terminus (aa 431), suggesting that CC1α1 pivoted around the CC1α1/α2 linker (*Figure 6A*, aa 242:431′).

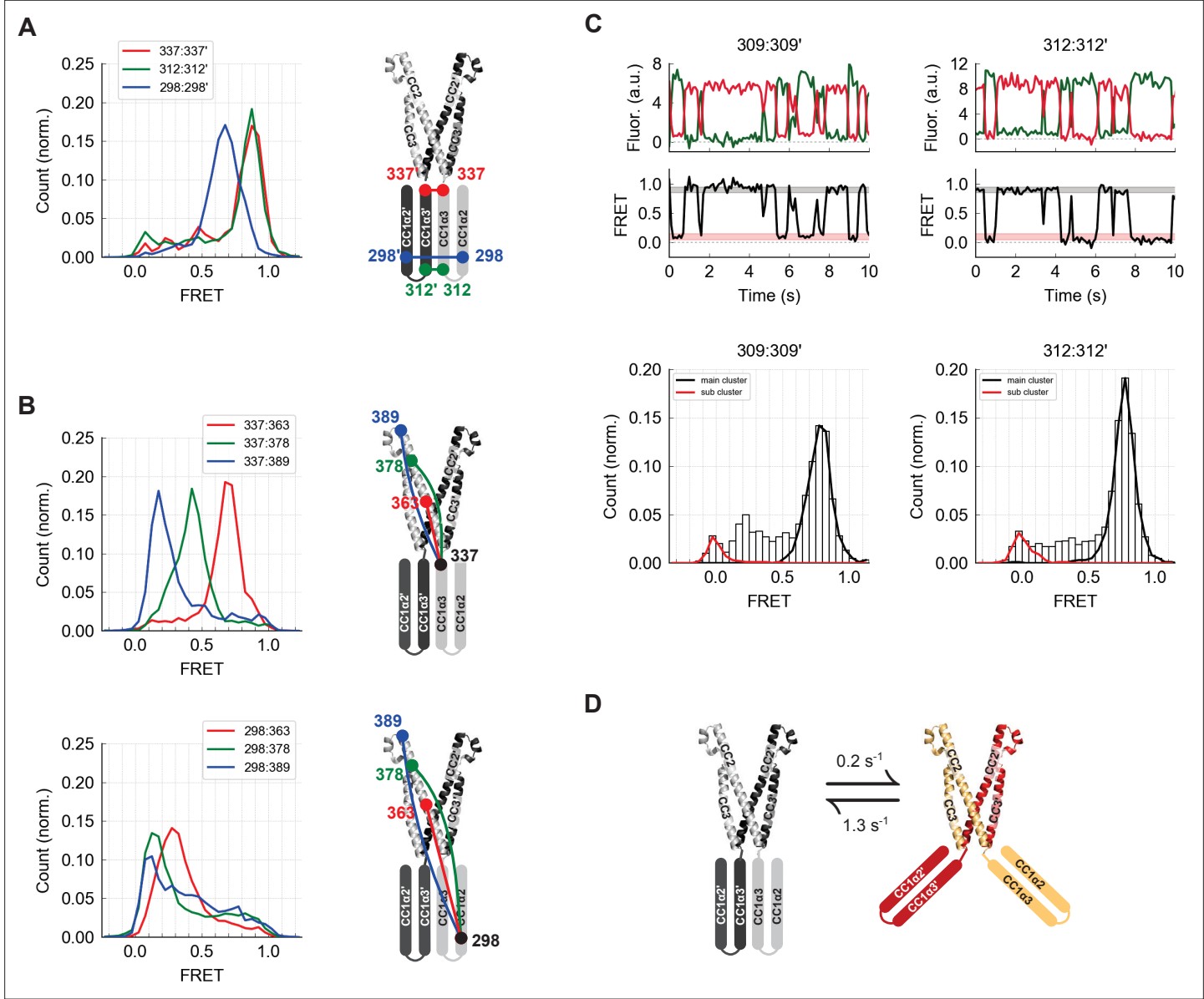

**Figure 4.** The CC1α2 and CC1α3 domains are closely apposed and directed away from CAD. (**A**) High inter-subunit smFRET at the N- and C-termini of the CC1α3 domain (aa 312 and 337, respectively) indicates close parallel apposition of the two helices. The crystal structure of CAD is shown with a schematic depiction of CC1α2 and CC1α3 domains. (**B**) (*top*) High intra-subunit smFRET from aa 337 to sites on CC2 declined progressively with distance, while intra-subunit smFRET from aa 298 (near the CC1α2/α3 linker) to sites in CC2 was low (*bottom*), indicating that the CC1α2 and α3 domains are directed away from CAD. (**C**) Inter-subunit recordings near the CC1α3 N-terminus displayed brief transitions from the predominant high-FRET state to a low-FRET state. Ensemble histograms show similar main and sub cluster FRET states for neighboring sites aa 309 and 312. (**D**) Schematic depiction of spontaneous conformational transitions of the CC1α3 domains. CC1α3 domains are closely apposed (*left*) but occasionally transition to a open configuration (*right*) at rates derived from dwell time analysis of smFRET traces (*Figure 4—figure supplement 1E,F*).

The online version of this article includes the following figure supplement(s) for figure 4:

**Figure supplement 1.** CC1α3:CC1α3′ dynamics.

**Figure supplement 1—source data 1.** Time series of FRET, Hidden-Markov Model fits, and state clusters for *Figure 4—figure supplement 1*.

To determine how R304W releases CC1α1 given its distant location at the C-terminal end of CC1α2, we examined its effect on folding of the CC1α2 and CC1α3 domains. Intra-subunit smFRET measurements showed that R304W increases the separation between the CC1α2 N-terminus and the CC1α3 C-terminus (*Figure 6A*, aa 274:337), suggesting it has opened the tight angle between CC1α2

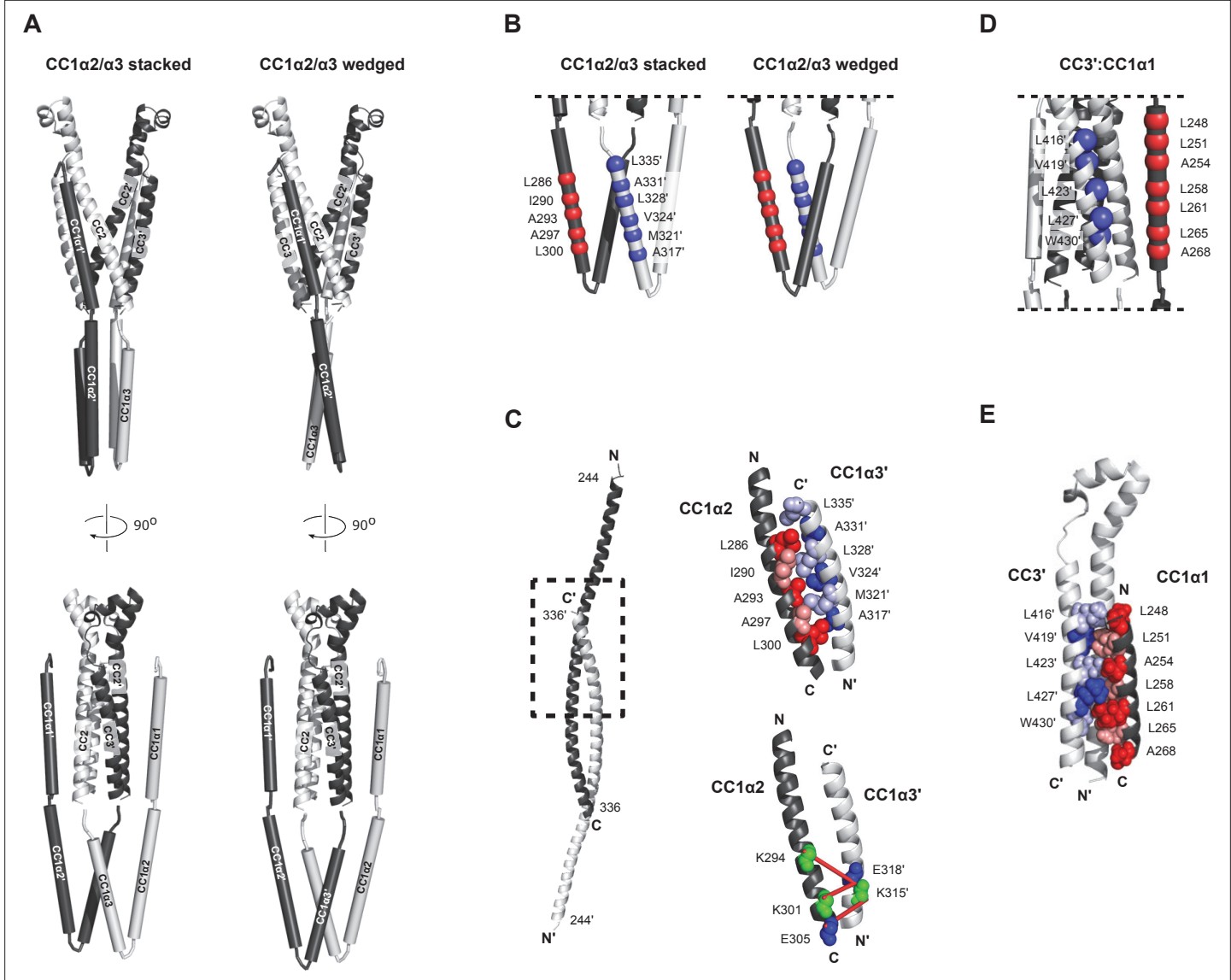

**Figure 5.** smFRET-derived models of the configuration of CC1 and CAD in ctSTIM1. (**A**) 36 smFRET-derived distances were used to reconstruct the orientations of CC1 domains relative to CAD, yielding two classes of solutions with the CC1α2/α3 domains in a 'stacked' (*left*) or 'wedged' (*right*) configuration. In both classes, CC1α1 domains are in close parallel apposition to the CC3 domains on the adjacent subunit in a domain-swapped configuration, with the CC1α2 and CC1α3 domains forming a compact parallel structure directed away from CAD. The average of 50 model solutions is shown for each class (see *Figure 5—figure supplement 1* for individual solutions). The complete list of smFRET values and distance constraints used to generate the models is shown in *Supplementary file 1*. (**B**) smFRET-derived models suggest hydrophobic stabilization of the CC1α2/α3 complex by antiparallel apposition of CC1α2 and CC1α3'. (**C**) The crystal structure of CC1 peptides depicts an antiparallel interaction of CC1α2 and CC1α3' domains (*left*, *dashed box*), in which hydrophobic residues form a tightly packed dimer interface (*top right*, adapted from 4O9B.pdb). (*Bottom right*) Tight antiparallel packing of CC1α2/α3' in ctSTIM1 was confirmed by inter-subunit crosslinks with EDC (*red lines*) (see also *Figure 5—figure supplement 3*). (**D**) Parallel apposition of hydrophobic residues on CC1α1 and CC3'. Many of these were previously identified by mutagenesis to stabilize the inactive state of STIM1, including L248, L251, L258, L261, L416, V419, and L423. (**E**) A model of the hydrophobic CC1α1:CC3' interface obtained by computational peptide docking (see Materials and methods).

The online version of this article includes the following figure supplement(s) for figure 5:

**Figure supplement 1.** CC1-CAD models.

**Figure supplement 2.** Mass spectrometry analysis of ctSTIM1 crosslinking by BS[3].

**Figure supplement 2—source data 1.** Raw unedited gel for *Figure 5—figure supplement 2A* (ctSTIM1 BS3 crosslinking).

**Figure supplement 2—source data 2.** Uncropped labeled gel for *Figure 5—figure supplement 2A* (ctSTIM1 BS3 crosslinking).

**Figure supplement 3.** Mass spectrometry analysis of ctSTIM1 crosslinking by EDC.

*Figure 5 continued on next page*

*Figure 5 continued*

**Figure supplement 3—source data 1.** Raw unedited gel for *Figure 5—figure supplement 3A* (ctSTIM1 EDC crosslinking).

**Figure supplement 3—source data 2.** Uncropped labeled gel for *Figure 5—figure supplement 3A* (ctSTIM1 EDC crosslinking).

and CC1α3. R304W also shifted CC1α2:CC1α2′ and CC1α3:CC1α3′ FRET to lower levels (*Figure 6B*), again consistent with opening of the compact inactive structure.

The R304W mutation also significantly altered the structural dynamics of CC1α2/α3. In the wild-type protein, the predominant inter-subunit FRET efficiency at the CC1α3 N-terminus was high, interrupted by occasional transitions to a low FRET state (*Figures 4C and 6C*, left). In contrast, the R304W mutation appeared to destabilize the CC1α3 pair, such that it transitioned between high and low FRET through a series of intermediates states (*Figure 6C*, right). Taken together, our data reveal several effects of R304W on CC1: it destabilizes the tight packing of CC1α3 domains, causes CC1α2 to splay out from CC1α3, and completely releases CC1α1 from CAD.

We also assessed effects of R304W on the structure of CAD. While the R304W mutation or the deletion of CC1 (ΔCC1) did not affect inter-subunit FRET efficiencies at the base of CAD (431:431′), they did alter the smFRET distributions at the CAD apex (389:389′ and 400:400′) (*Figure 6D*). Together, these results suggest that the association of CC1α1 with CAD influences the conformation of the flexible apex without affecting the arrangement of helices at the more stable CAD base.

## The CC1 domains refold into a parallel dimer during STIM1 activation

We examined cysteine crosslinking of the CC1 domains in ctSTIM1 to complement the structural information derived from smFRET. Inter-subunit cysteine crosslinking in ctSTIM1 was notably robust in the CC1α2/α3 linker (T307C) and the C-terminus of CC1α3 (S339C) (*Figure 7—figure supplement 1A*). While disulfide formation at T307C appeared consistent with the close apposition of CC1α3 domains implied by smFRET for the inactive ctSTIM1 conformation (*Figure 4*), further tests with BS³ crosslinking suggested these disulfides only form in the active state. Disulfide formation at T307C prevented BS³ from crosslinking CC1α1 and CAD, implying that T307C disulfides are incompatible with the CC1α1-CAD inhibitory clamp (*Figure 7—figure supplement 2D*). We observed a similar loss of BS³ crosslinks between CC1α1 and CAD in the activated ctSTIM1 mutants L248S/L251S (*Muik et al., 2011*) and R304W (*Figure 7—figure supplement 2B, C*), suggesting that disulfide formation at T307C occurred during spontaneous, transient visits to an activated state of ctSTIM1. Together, these results suggest that while CC1α3 domains are closely apposed in both the inactive and active conformations, only in the active conformation are T307C residues sufficiently close and properly oriented for disulfides to form.

Under physiological conditions, full-length STIM1 (flSTIM1) is activated by ER Ca²⁺ store depletion, and the CC1 domains are anchored to the ER membrane. To extend our in vitro ctSTIM1 results to flSTIM1 in cells, we examined cysteine crosslinking of CC1 domains in inactive and active flSTIM1 at sites that we had studied in vitro (H266, T274, T307, N309, and S339) as well as the distal region of CC1α1 (aa 262–269) immediately beyond the region in CC1α1 thought to dimerize upon activation (aa 233–261) (*Hirve et al., 2018*). Cysteine mutants were transiently expressed in HEK293 cells and treated with the cell-permeant oxidizer diamide before or after store depletion induced by cyclopiazonic acid (CPA). Lysates of diamide-treated cells were then analyzed for the presence of crosslinked flSTIM1 dimers. Prior to store depletion, when flSTIM1 is inactive, little crosslinking occurred throughout CC1, with the exception of S339C near the junction with CAD (*Figure 7A and B*). In contrast, after store depletion prominent crosslinking occurred at A268C and T307C in addition to S339C (*Figure 7A and B*, and *Figure 7—figure supplement 1B*). These results further support our conclusion that T307C crosslinking of ctSTIM1 in vitro is enabled by transient activation events. More importantly, they suggest that activation in vivo brings all three CC1 subdomains together.

To test this idea, we asked whether cysteine crosslinking after ER Ca²⁺ store depletion could lock flSTIM1 in the activated state, thereby preventing the deactivation of STIM1 and SOCE that normally occurs upon store refilling. Ionomycin was added in Ca²⁺-free Ringer's to deplete stores, after which ionomycin was removed and Ca²⁺ readded to allow Ca²⁺ influx through SOCE. In HEK cells expressing WT flSTIM1, [Ca²⁺]ᵢ increased rapidly upon Ca²⁺ readdition and subsequently declined due to store refilling and deactivation of STIM1 and Orai1, regardless of diamide treatment (*Figure 7C*). We obtained a similar result with flSTIM1-S339C, indicating that diamide-induced crosslinking at S339C

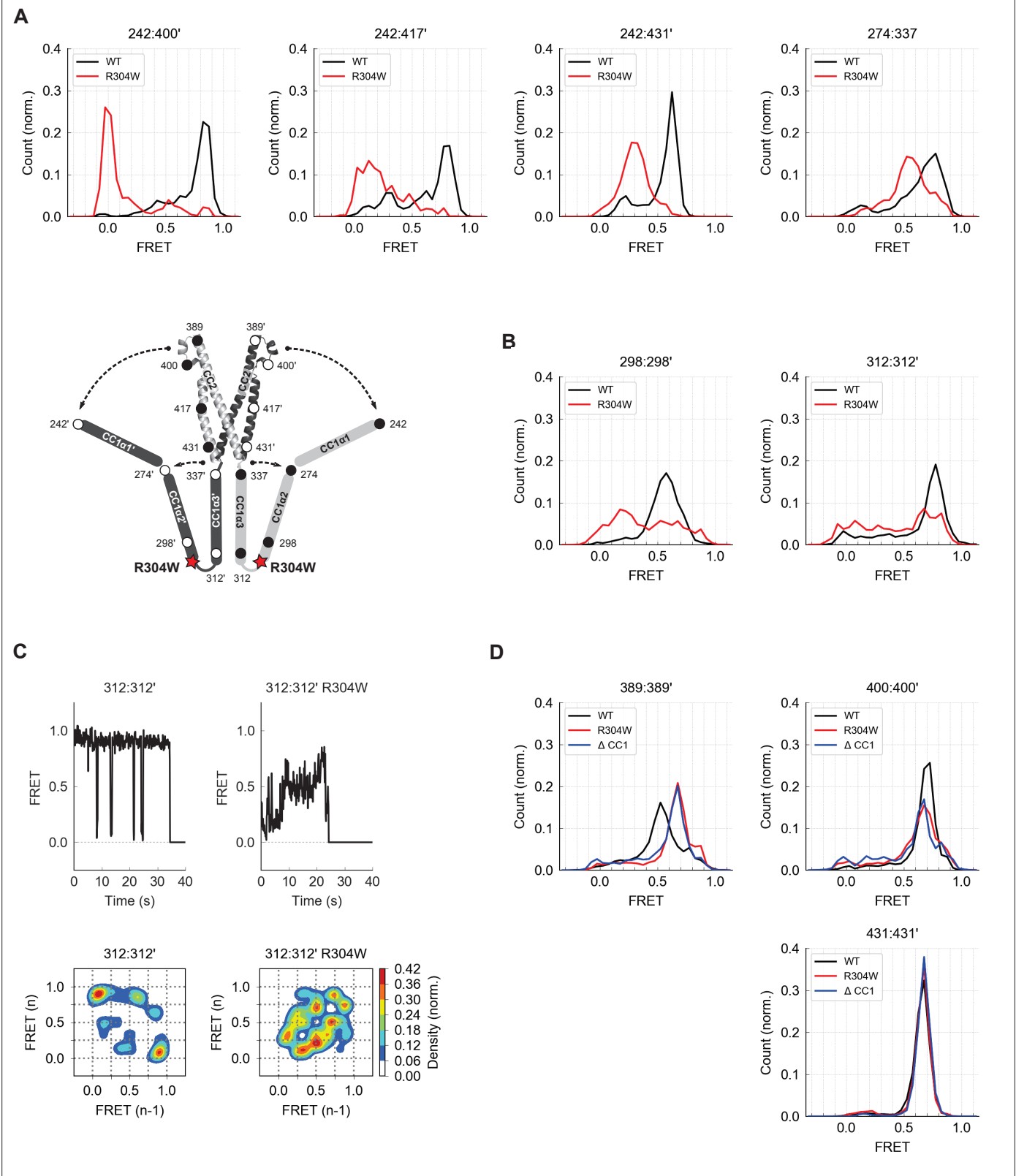

**Figure 6.** The Stormorken R304W mutation releases CC1 from CAD. (**A**) The R304W mutation reduced inter-subunit smFRET between aa 242 on CC1α1 and sites 400′, 417′, and 431′ on CC3′, indicating separation of CC1α1 from CAD. The mutation also reduced intra-subunit FRET for 274:337, consistent with a widening of the angle between CC1α2 and CC1α3. (**B**) The predominant FRET levels for 298:298′ (*left*) and 312:312′ (*right*) were diminished by R304W, suggesting a disruption of the compact parallel structure of CC1α2/α3 domains. (**C**) Effects of R304W on inter-subunit FRET fluctuations at

*Figure 6 continued on next page*

*Figure 6 continued*

312:312′, near the CC1α3 N-terminus. The dominant large-amplitude fluctuations of the wild-type (*top left*) were entirely absent in the mutant, which instead displayed frequent small fluctuations among a range of FRET levels (*top right*). The different fluctuation modes are summarized in ensemble transition density plots (see Materials and methods) which show a single dominant FRET transition for wild-type molecules (*bottom left*, 38 molecules), and a multimodal density distribution for R304W mutants (*bottom right*, 25 molecules). (**D**) In the CAD apex, R304W or deletion of the entire CC1 region (ctSTIM1-ΔCC1) increased the FRET at 389:389′ (*left*) and reduced it slightly at 400:400′ (*center*), suggesting that the apex interacts with CC1α1 in the WT conformation. Neither R304W nor ΔCC1 affected FRET at 431:431′ (*right*) in the CAD base.

The online version of this article includes the following figure supplement(s) for figure 6:

**Figure supplement 1.** Orai1-mediated Ca$^{2+}$ influx evoked by ctSTIM1 fragments.

does not impede flSTIM1 deactivation. In contrast, crosslinking at A268C and T307C prevented the normal deactivation of SOCE as indicated by the sustained elevation of [Ca$^{2+}$]$_i$ after Ca$^{2+}$ readdition. These findings suggest that all three CC1 subdomains are brought close together during STIM1 activation in vivo and remain associated in the final, Orai1-bound conformation (**Figure 7D**).

## Discussion

Precise regulation of SOCE is essential for many physiological processes and is achieved through control of transitions between inactive and active states of STIM1. These transitions are triggered by changes in ER [Ca$^{2+}$] that toggle reversible interactions of the STIM1 CC1 domain with CAD. Through a combination of smFRET measurements and crosslinking assays, we have determined the arrangements of CC1 and CAD helices in inactive and active states of STIM1. The results reveal how CC1 is oriented around CAD to stabilize the inactive state, and how its conformation changes upon activation to project CAD toward the PM to engage Orai1 and trigger SOCE.

In this study, we measured FRET efficiencies from single ctSTIM1 dimers to generate an overlapping set of intramolecular distance constraints for modeling tertiary structure (**Choi et al., 2010**; **Brunger et al., 2011**). Caution must be applied in using FRET signals to infer absolute distances, as FRET efficiency can also be influenced by photophysical transitions or interactions with the local environment (**Weiss, 1999**; **Weiss, 2000**). We employed several strategies to avoid misinterpretation of the data. First, by including a large number of different FRET pairs (36 pairs for CC1-CAD), no major feature of the CC1-CAD model was dependent on any single measurement, safeguarding against potential problems with individual label sites (**Figure 5—figure supplement 1A**). Second, we independently verified the CC1-CAD model by mass spectrometry analysis of BS$^3$-crosslinked protein (**Schmidt and Robinson, 2014**; **Figure 5—figure supplement 2**). Third, measurements of key smFRET amplitudes and spontaneous structural transitions were confirmed at multiple nearby sites (**Figure 2** and **Figure 4—figure supplement 1**). Finally, we applied relatively conservative distance error limits in constraining the CC1-CAD model (**Supplementary file 1**). Taken together, these strategies help validate the tertiary structures represented in our models.

### The conformation of CAD in the context of ctSTIM1

smFRET measurements demonstrated that CAD within the context of ctSTIM1 resembles the CAD crystal structure rather than the NMR structure of CC1α3-CC2 (**Figure 1**). The base is stable and compact (**Figure 2**), as expected given the hydrophobic interactions and hydrogen bonding between the two CAD subunits in this region (**Yang et al., 2012**). Surprisingly, the CAD apex appeared flexible and diverged from the crystal structure (**Figure 2**). The flexibility of the CAD apex is intriguing because mutagenesis and cysteine-crosslinking studies suggest this region interacts with Orai1 to initiate channel activation (**Calloway et al., 2010**; **Korzeniowski et al., 2010**; **Wang et al., 2014**; **Thompson et al., 2018**; **Butorac et al., 2019**), and conformational flexibility can facilitate protein-protein interactions through effects on the thermodynamics of binding (**Grünberg et al., 2006**). Thus, it seems likely that the CAD apex in complex with Orai adopts a structure different from that depicted in the CAD crystal and distinct from the NMR structure of CC1α3-CC2 (see below). Release of CC1α1 from CAD, or deletion of the CC1 domain altogether, did not cause a gross change in CAD structure, although it did alter the conformation of the CAD apex (**Figure 6**). These results raise the possibility that upon CC1α1 dissociation the apex changes shape, perhaps facilitating its binding to Orai1.

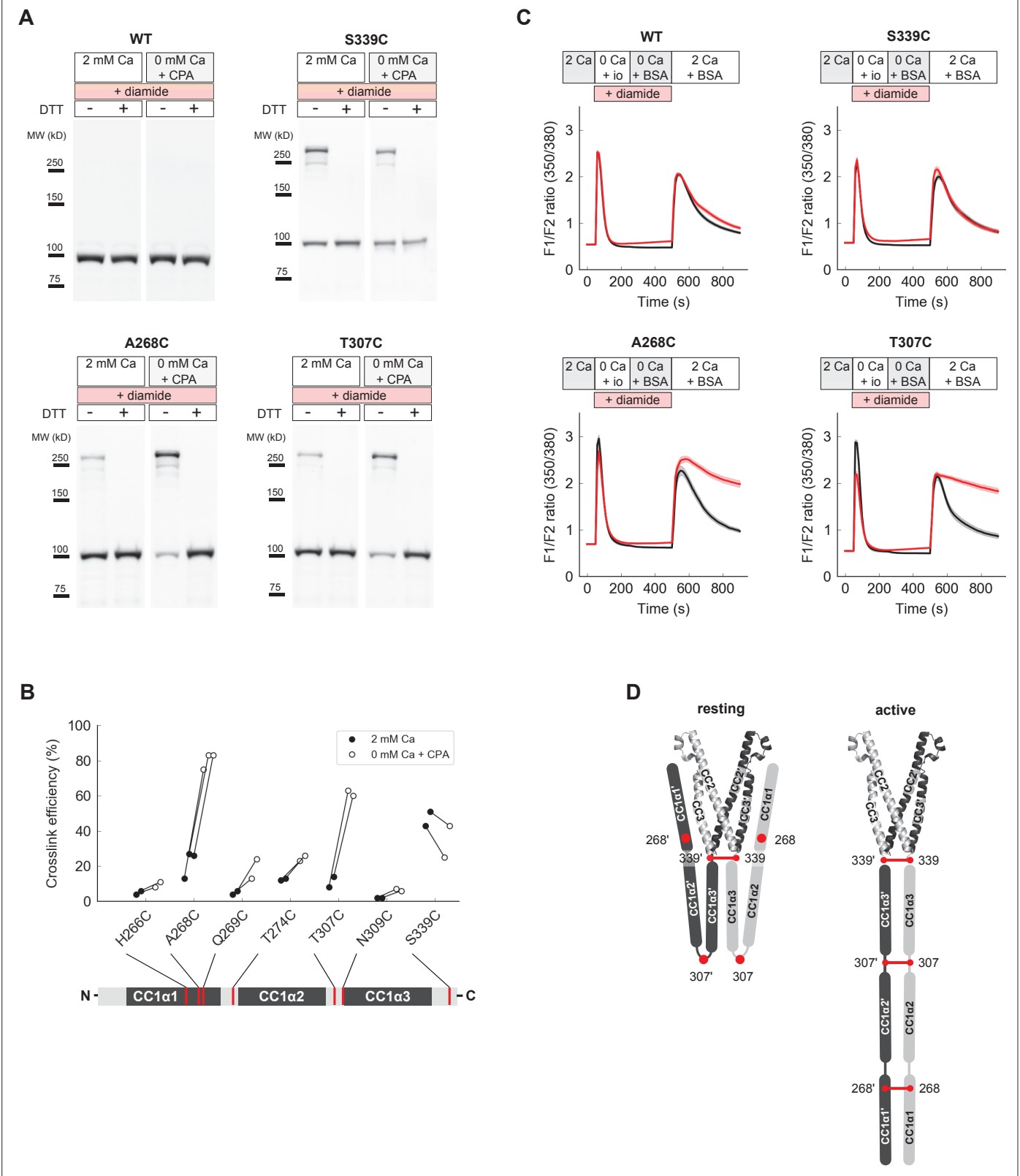

**Figure 7.** Close pairing of CC1 domains along their entire length in the activated state of full-length STIM1 (flSTIM1) in vivo. (**A**) Western-blot analysis of diamide-induced cysteine crosslinking of flSTIM1-WT, flSTIM1-A268C, flSTIM1-T307C and flSTIM1-S339C in HEK293 cells, under resting (2 mM Ca²⁺) or store-depleted (0 mM Ca²⁺ + CPA) conditions. (**B**) Summary of flSTIM1 cysteine crosslinking before (*black*) and after (*white*) store depletion measured in individual paired experiments. While crosslinking at aa 268 and aa 307 strongly increased in the activated state, crosslinking at aa 339 occurred

*Figure 7 continued on next page*

*Figure 7 continued*

independently of STIM1 activation (see also *Figure 7—figure supplement 1B*). (**C**) Effects of flSTIM1 cysteine crosslinking on deactivation of SOCE following store refilling. WT flSTIM1 and cysteine mutants were co-expressed with Orai1 for cytosolic calcium imaging. In cells expressing WT flSTIM1 and store-depleted by transient exposure to ionomycin (io, 1 µM), addition of 2 mM $Ca^{2+}$ elevated $[Ca^{2+}]$, due to SOCE, followed by a decline as SOCE deactivated from store refilling (*top left, black*). In contrast, diamide-induced crosslinking of A268C or T307C flSTIM1 mutants stabilized the active state, as evidenced by persistent calcium influx after ionomycin wash-out and store refilling (*bottom left and right, red*). Crosslinking of S339C did not affect deactivation of SOCE upon store refilling (*top right*). Each trace shows the mean and s.e.m. of the following numbers of cells (control/diamide) from at least two independent experiments: WT (91/106), A268C (31/47), T307C (38/46), S339C (44/41). (**D**) Schematic illustration of CC1 cysteine crosslinking in the resting (*left*) and activated (*right*) states of flSTIM1 (only CC1 and CAD are shown for clarity). In the resting state, CC1α1 and CC1α3 domains are kept apart, preventing crosslinking at locations upstream of aa 339. Upon store depletion, release of the CC1α1 domains from CAD promotes alignment of CC1 domains along their entire length, enabling crosslinking at aa 268 and 307.

The online version of this article includes the following figure supplement(s) for figure 7:

**Source data 1.** Raw unedited and uncropped labeled western blots for *Figure 7A* (WT).

**Figure supplement 1.** CC1 cysteine crosslinking in ctSTIM1 and flSTIM1.

**Figure supplement 1—source data 1.** Raw unedited and uncropped labeled gel and western blots for *Figure 7—figure supplement 1*.

**Figure supplement 2.** Mass spectrometry analysis of BS³ crosslinking of active ctSTIM1 mutants.

**Figure supplement 2—source data 1.** List of crosslinked residues from mass spectrometry.

## Defining structural roles for all three CC1 subdomains in the STIM1 inhibitory clamp

Our results reveal how the three helical regions within CC1 collaborate to regulate the activity of STIM1. Prior evidence indicated that CC1 autoinhibits STIM1 in resting cells, but fundamental questions concerning its structure remained. Korzeniowski and colleagues originally reported that neutralization of acidic residues in the CC1α3 domain activated STIM1 and proposed an 'inhibitory clamp' mechanism in which this region binds to basic residues in CAD to shield it from Orai in resting cells (*Korzeniowski et al., 2010*). However, subsequent studies showed that CC1α3 alone is not sufficient to maintain the inactive state, as CC1α3-CAD is fully active (*Zhou et al., 2013*), and deletion of CC1α3 in flSTIM1 only slightly activated Orai1 current under resting conditions (*Fahrner et al., 2014*). Moreover, inter-molecular FRET between ER-tethered STIM1 fragments showed that CC1α1 can bind to CC3 (*Fahrner et al., 2014*), and ER-tethered CC1α1 by itself can sequester CAD and prevent it from binding to Orai1 in the PM (*Fahrner et al., 2014*; *Ma et al., 2015*). These findings and the activating effects of mutations in CC1α1 (*Muik et al., 2011*; *Zhou et al., 2013*; *Fahrner et al., 2014*; *Ma et al., 2015*) and CC3 (*Muik et al., 2011*) shifted the focus to a CC1α1-CC3 interaction as the basis for the inhibitory clamp.

Based largely on mutagenesis effects and helical wheel modeling, Zhou and colleagues proposed that the inhibitory clamp arises from antiparallel binding of CC1α1 to CC3, involving interactions between L261 and L416 and between L258 and V419 (*Ma et al., 2015*). Our FRET measurements support a different arrangement in which CC1α1 binds parallel to CC3 (*Figure 3*), and our model depicts direct interactions of L248 and L251 (CC1α1) with L416 (CC3), and of L258 and L261 (CC1α1) with L423 and L427 (CC3) (*Figure 5*). Such a CC1α1-CC3 interface may explain the activating effects of non-conservative mutations at these sites (L248S, L251S, L416G, L258G/A, L261G, L423G) (*Muik et al., 2011*; *Zhou et al., 2013*; *Fahrner et al., 2014*; *Ma et al., 2015*). Interestingly, our data reveal that CC1α1 preferentially associates with CC3 of the adjacent subunit in the CAD dimer (*Figure 3*). This domain-swapped configuration may enhance the stability of the inactive state through inter-subunit interactions and could promote cooperativity between subunits during structural transitions.

The CC1 helices displayed large spontaneous FRET fluctuations, indicating a high degree of flexibility. The CC1α2/α3 domains occasionally transitioned from a compact, high-FRET resting state to a low-FRET open state (*Figure 4*), while the CC1α1 domains briefly switched sides on CAD, probably trading binding interfaces (*Figure 3*). Differences in dwell times suggest these two types of fluctuations occurred independently (*Figure 4—figure supplement 1G*), but further studies will be needed to determine how they relate to structural changes during activation of membrane-inserted flSTIM1.

A recent report described a solution NMR structure of the isolated CC1 peptide, in which the CC1α1/α2/α3 helices form a monomeric three-helix bundle with an extensive coiled-coil interaction between CC1α1 and CC1α2 (*Rathner et al., 2021*). This arrangement is quite distinct from our

FRET-derived model which features prominent intersubunit interactions of CC1α1 with CC3, and of CC1α2 with CC1α3. There are several technical differences between the two studies that could contribute to this discrepancy. In the NMR study, CC1 was studied in isolation with 7 mM SDS to prevent oligomerization, thus depriving CC1α1 of its normal CAD-binding partner and discouraging intersubunit CC1:CC1′ binding. While it is possible these conditions promoted formation of a structure that does not exist under physiological conditions, a more intriguing possibility is that the monomeric CC1 structure represents a pre-activated state of CC1 in which the interaction of CC1α1 with CC1α2 has destabilized the CC1α1:CC3′ clamp (*Rathner et al., 2021*). Passage through such an intermediate bundled state could serve to mitigate the energetic cost of breaking the CC1α1:CC3′ and CC1α2:CC1α3′ interactions and thereby ease the transition between the inactive and active states. Our smFRET experiments show that when CC1α1 associates with CC3 in ctSTIM1, the CC1α2 and CC1α3 domains primarily interact with each other to form a compact bundle pointing away from CAD (*Figure 5*); in the context of the Rathner et al report, the spontaneous fluctuations of CC1α2/α3 we observed (*Figure 4C and D*) may represent spontaneous transitions towards such a pre-activated state. smFRET studies of flSTIM1 may offer an effective approach for identifying intermediate conformational states during the activation process.

The R304W Stormorken mutation activates ctSTIM1 (*Figure 6—figure supplement 1*), consistent with its ability to activate flSTIM1 (*Misceo et al., 2014*; *Morin et al., 2014*; *Nesin et al., 2014*; *Fahrner et al., 2018*). smFRET measurements show that R304W destabilizes the compact packing of CC1α2/α3 domains, increasing the angle between CC1α2 and CC1α3 helices and fully releasing CC1α1 from CAD (*Figure 6*). These results are consistent with NMR studies showing that R304W causes N-terminal extension of the CC1α3 helix, thereby stiffening the CC1α2/α3 linker (*Fahrner et al., 2018*; *Rathner et al., 2021*). Release of CC1α1 from CAD alone may be sufficient to explain the activating effects of R304W on ctSTIM1; smFRET values for 242:242′ and 298:298′ were low, suggesting that the CC1 helices remain separated in ctSTIM1 R304W. In the context of flSTIM1 coiled-coil formation by the transmembrane domains may stabilize paired association of the CC1 helices following store depletion (*Hirve et al., 2018*). It is not known whether R304W fully mimics these effects of store depletion on the tertiary structure of CC1 in flSTIM1.

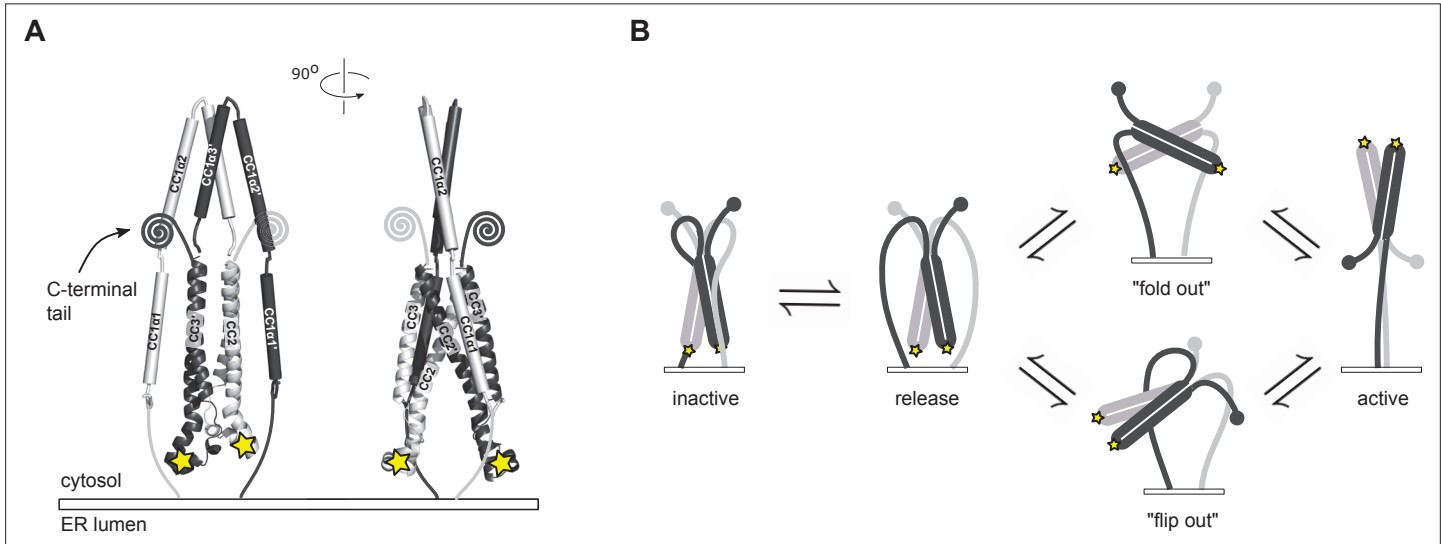

**Figure 8.** Possible conformational trajectories of flSTIM1 activation in vivo. (**A**) In the resting state, the parallel orientation of CC1α1 and CC3 domains implies that CAD is held with its apex close to and pointed towards the ER membrane, effectively shielding critical interaction sites (*stars*) from engaging with Orai. The region C-terminal to CAD is abbreviated as its conformation is unknown. (**B**) Alternative models for transitions to the active state of flSTIM1. In the 'fold out' model (*top*), following the release of CC1α1, CAD undergoes a symmetric conformational change in which the CC2 domains pass through a transient antiparallel state, providing a free path for the region downstream of CAD to extend towards the plasma membrane. In the 'flip out' model (*bottom*), CAD maintains its parallel resting conformation and undergoes an asymmetric outward rotation from between the CC1 domains. Assuming that the initial resting conformation is symmetric, one of its C-terminal domains (*gray*) would need to be carried through the gap between the two CC1 domains.

## A structural view of quiescent STIM1 and its transition to the activated state

Our results provide a first view of how the CC1 domain of STIM1 sequesters CAD near the ER membrane in resting cells to prevent it from interacting with Orai. The parallel arrangement of CC1α1 and CC3 in our smFRET-derived structure predicts that the CAD apex of inactive STIM1 is juxtaposed to the ER membrane and pointed away from the PM in vivo (*Figure 8A*). This orientation implies that to activate STIM1 and SOCE, the CAD apex must rotate through a large angle (up to 180°) and translocate more than 10 nm toward the PM - a massive conformational change reminiscent of the refolding and extension of viral envelope proteins like hemagglutinin when stimulated to trigger membrane fusion (*Harrison, 2008*). Hirve et al. showed that activation of flSTIM1 is initiated by formation of a coiled-coil in the TM domains and the proximal part of CC1α1 up to L251, with a high probability of coiled-coil assembly predicted through A268 (*Hirve et al., 2018*). As an extension to these findings, we found that after store depletion, inter-subunit disulfide bonds form at A268C (C-terminus of CC1α1) and T307C and S339C (N- and C-termini of CC1α3, respectively; *Figure 7*). Taken together, these results suggest that all three helical domains in CC1 are closely paired in the active dimer, which would effectively move CAD across the ~15 nm gap of the ER-PM junction to reach Orai (*Wu et al., 2006*).

Our CC1-CAD model specifies the possible types of conformational changes that must occur to activate flSTIM1 and present the CAD apex to Orai (*Figure 8B*). We can envision two potential solutions. In a 'flip-out' mechanism, CAD rotates by 180°. Although our smFRET data indicate that release of CC1α1 does not cause a gross conformational change in CAD (*Figure 6*), this kind of CAD rotation would imply a complex asymmetric intermediate, in which one of the C-terminal domains downstream of CAD (aa 449–685) would need to be threaded between the two CC1 domains. Alternatively, in a 'fold-out' model, CAD undergoes a symmetric conformational change while moving its apex away from the ER membrane. This model posits that during activation the CC2 domains rotate to transiently assume an antiparallel state, reminiscent of the NMR structure (*Figures 1A and 8B*). It is important to note that our cysteine crosslinking results indicate the CC1α3 C-termini are closely apposed in the Orai-bound state (*Figure 7D*, S339C), which is incompatible with the splayed configuration of CC1α3 domains in the NMR structure. Thus, if the antiparallel conformation exists, it is likely to represent a transient intermediate during the activation of STIM1 (*Stathopulos et al., 2013*).

The ultimate endpoint in delineating the mechanism of SOCE is an atomic-level description of the pathway of STIM1 activation by store depletion. To reach this goal the starting, intermediate, and final conformations of STIM1 after store depletion must be resolved, as well as the structure of the activated STIM-Orai complex. While approaches like cryo-EM certainly offer an effective strategy for elucidating structures, time-resolved measurements such as those afforded by smFRET will be essential to identify structural transitions in the activation pathway. The results obtained thus far through smFRET of ctSTIM1 have revealed critical measurement sites that offer a starting point for these studies. Ultimately, the identification of transient intermediate STIM1 conformations and rate-limiting steps in the activation process may reveal critical transitions that could be targeted to manipulate SOCE for therapeutic benefit.

# Materials and methods

**Key resources table**

| Reagent type (species) or resource | Designation | Source or reference | Identifiers | Additional information |
|---|---|---|---|---|
| Gene (*human*) | STIM1 | Origene | | |
| Strain, strain background (*Escherichia coli*) | BL21 | New England Biolabs | C25271 | |
| Strain, strain background (*Escherichia coli*) | CVB101 | Avidity | CVB101 | |
| Cell line (*human*) | HEK293 | ATCC | CRL-1573 | |
| Antibody | Mouse monoclonal anti-mCherry | Takara Bio | 632,543 RRID:AB_2307319 | (1:2000) |

*Continued on next page*

*Continued*

| Reagent type (species) or resource | Designation | Source or reference | Identifiers | Additional information |
|---|---|---|---|---|
| Antibody | Donkey polyclonal anti-mouse IgG | LI-COR | 926–032212 RRID:AB_621847 | (1:10000) |
| Recombinant DNA reagent | pAC6 (vector) | Avidity | | |
| Recombinant DNA reagent | mCherry-labeled STIM1 (plasmid) | doi:10.1083/jcb.200604014 | | |
| Recombinant DNA reagent | Orai1-GFP (plasmid) | doi:10.1074/jbc.M703573200 | | |
| Peptide, recombinant protein | TEV protease | MCLAB | TEV-200 | |
| Peptide, recombinant protein | BSA-biotin | Sigma Aldrich | A8549 | |
| Peptide, recombinant protein | Neutravidin | Thermo Fisher | 31000 | |
| Peptide, recombinant protein | Glucose oxidase | Sigma Aldrich | G2133 | |
| Peptide, recombinant protein | Catalase | Sigma Aldrich | C9322 | |
| Peptide, recombinant protein | Trypsin/LysC protease | Promega | V5071 | |
| Commercial assay or kit | QuikChange II | Agilent | 200524 | |
| Commercial assay or kit | PEG/PEG-biotin | Laysan Bio | BIO-PEG-SVA-5K & MPEG-SVA-5K | |
| Chemical compound, drug | TCEP | Thermo Fisher Scientific | 77720 | |
| Chemical compound, drug | Alexa Fluor 555 | Invitrogen Life Technologies | A-20346 | |
| Chemical compound, drug | Alexa Fluor 647 | Invitrogen Life Technologies | A-20347 | |
| Chemical compound, drug | Biotinylated lipids 18:1 Biotinyl Cap PE | Avanti Polar Lipids | 870273 C | |
| Chemical compound, drug | egg-PC lipids | Avanti Polar Lipids | 840051 C | |
| Chemical compound, drug | Cyclooctatetraene | Sigma Aldrich | 138924 | |
| Chemical compound, drug | $BS^3$-d0 | Sigma Aldrich | 21590 | |
| Chemical compound, drug | $BS^3$-d4 | Sigma Aldrich | 21595 | |
| Chemical compound, drug | EDC | Thermo Fisher Scientific | 77149 | |
| Chemical compound, drug | Sulfo-NHS | Thermo Fisher Scientific | 24520 | |
| Chemical compound, drug | ProteaseMax | Promega | V2071 | |
| Chemical compound, drug | Cyclopiazonic acid (CPA) | Sigma Aldrich | C1530 | |
| Chemical compound, drug | Protease inhibitor cocktail | Cell Signaling Technology | 5871 S | |
| Chemical compound, drug | Lipofectamine 2000 | Thermo Scientific | 11668027 | |
| Chemical compound, drug | Poly-D-lysine | Sigma Aldrich | P-7405 | |
| Chemical compound, drug | fura-2/AM | Invitrogen | F-1221 | |
| Software, algorithm | µManager | doi:https://doi.org/10.1002/0471142727 | RRID:SCR_016865 | https://micro-manager.org/ |
| Software, algorithm | SMART: Single Molecule Analysis Research Tool | doi:10.1371/journal.pone.0030024 | | https://simtk.org/projects/smart/ |

*Continued on next page*

*Continued*

| Reagent type (species) or resource | Designation | Source or reference | Identifiers | Additional information |
|---|---|---|---|---|
| Software, algorithm | Crystallography and NMR System (CNS) | doi:10.1038/nprot.2007.406; doi:https://doi.org/10.1107/S0907444998003254 | RRID:SCR_014223 | http://cns-online.org/v1.2/ |
| Software, algorithm | PyMOL Molecular Graphics System ver. 1.8 | Schrödinger, LLC | RRID:SCR_000305 | http://www.pymol.org/ |
| Software, algorithm | ProDy | doi:10.1093/bioinformatics/btr168 | | http://prody.csb.pitt.edu/ |
| Software, algorithm | Flexpepdock server of Rosetta | doi:10.1093/nar/gkr431 | Rosetta, RRID:SCR_015701 | http://flexpepdock.furmanlab.cs.huji.ac.il/ |
| Software, algorithm | Byonic v2.12.0 or v2.14.27 | Protein Metrics | RRID:SCR_016735 | |
| Other | NiNTA beads | Qiagen | 30210 | |
| Other | SnakeSkin | Life Technologies | 68700 | |
| Other | HiTrap Q column | GE Healthcare Life Sciences | 17-1153-01 | |
| Other | Quartz microscope slides | G. Finkenbeiner Inc. | | For smFRET |
| Other | Microscope coverslips | Erie Scientific | 24 × 40 1.5 001 | For smFRET |
| Other | Liposome Extruder Set | Avanti Polar Lipids | 610023 | |
| Other | PC Membranes 0.1 μm | Avanti Polar Lipids | 610005-1EA | |
| Other | Sepharose CL-4B column | Sigma Aldrich | CL4B200 | |
| Other | OBIS 532 nm LS 150 mW laser | Coherent | 1280719 | For smFRET |
| Other | OBIS 637 nm LX 140 mW laser | Coherent | 1196626 | For smFRET |
| Other | Dichroic beamsplitter | Semrock | FF652-Di01−25 × 36 | For smFRET |
| Other | Donor band-pass filter | Semrock | FF01-580/60-25-D | For smFRET |
| Other | Acceptor band-pass filter | Semrock | FF01-731/137-25 | For smFRET |
| Other | Emission beam splitter | Cairn Research | OptoSplit-II | For smFRET |
| Other | EM-CCD camera | Andor | iXon DU897E | For smFRET |
| Other | Polychrome II | TILL Photonics | | For calcium imaging |
| Other | emission filter | Semrock | FF02-534/30-25 | For calcium imaging |
| Other | emission filter >480 nm | Chroma Technology Corp. | | For calcium imaging |
| Other | Flash4.0 sCMOS camera | Hamamatsu Corp. | | For calcium imaging |

## Cell lines and cell culture

HEK293 cells were authenticated and verified to be mycoplasma-free by PCR testing (ATCC). Cells were passaged in DMEM containing 10% FBS, 2 mM L-alanyl-glutamine, and 100 U/ml penicillin/streptomycin and cultured at 37°C in 5% $CO_2$.

## DNA constructs

For samples that were encapsulated in liposomes for surface immobilization, DNA encoding ctSTIM1 (aa 233–685) was amplified by PCR from full-length human STIM1 (Origene), appending an N-terminal NcoI cleavage site, and a C-terminal TEV protease recognition sequence (SENLYFQG) followed by a HindIII cleavage site. The ctSTIM1 insert contained a silent T1764C mutation in H588 to remove an endogenous NcoI site, and a G1310C mutation to replace the endogenous cysteine (C437S). ctSTIM1 inserts were ligated into the pET28a vector which encoded a C-terminal 6-His tag. ctSTIM1 cysteine mutants were created by site-directed mutagenesis (QuikChange II, Agilent). The translated ctSTIM1 protein retained the sequence SENLYFQ at the C terminus after cleavage of the 6-His tag by TEV.

For samples that were directly attached to a PEG-coated surface, the pAC6 vector was used with a C-terminal AviTag sequence GLNDIFEAQKIEWHE (Avidity). ctSTIM1 was first ligated into a pTEV5 vector which encoded a TEV-cleavable N-terminal 6-His tag. DNA encoding the 6-His tag, TEV recognition site and ctSTIM1 was then amplified by PCR, appending an XhoI cleavage site to the N terminus, and a HindIII cleavage site to the C terminus for ligation into the pAC6 vector. The resulting translated ctSTIM1 protein retained the N-terminal sequence GAS after TEV cleavage of the 6-His tag, and it had the AviTag sequence following a spacer KLPAGG on the C terminus. All constructs were verified by DNA sequencing.

For cysteine mutagenesis and dye labeling we selected serine or threonine residues (structurally similar to cysteine) or polar residues (more likely to face the aqueous environment). In CC2, we chose residues that were expected to extend out into solution based on the NMR and crystal structures (*Figure 1A*). In CC1, we initially targeted predicted unstructured regions, in order to minimize interference with local secondary structure. Overall, we attempted to establish a reasonably even distribution of measurement sites throughout CC1 and CAD, although for certain regions of interest we selected several closely spaced residues for additional verification.

## Protein expression, purification, and labeling

ctSTIM1 protein was expressed in *E. coli* based on standard methods described previously (*Choi et al., 2010*). Briefly, plasmid containing the ctSTIM1 insert was transformed to BL21 competent cells (New England Biolabs) for pET28a plasmids, or CVB101 competent cells (Avidity) for pAC6 plasmids. Protein expression was induced with IPTG at OD ~0.4 in 450 ml Luria Broth (LB), and for pAC6 plasmids biotin was added to a concentration of 50 µM. Bacteria were spun down and then resuspended in denaturing PBS (8 M urea, 100 mM $NaH_2PO_4$, 10 mM Tris, pH 7.4 with NaOH) and lysed by sonication. TCEP (Thermo Fisher Scientific) was added to a concentration of 0.5 mM. Samples were purified by NiNTA purification (Qiagen) and subsequently slowly dialysed using SnakeSkin (Life Technologies #68700) to 20/50 TBS (50 mM NaCl, 20 mM Tris, pH 7.4 with HCl) containing 1 mM DTT. TEV protease (10 µg/ml, MCLAB) was added during dialysis. Cleavage of the 6-His tag was usually almost 100 % as checked by SDS-PAGE. Samples were further purified by ion exchange on a HiTrap Q column (GE Healthcare Life Sciences) and eluted in 250 µl aliquots of 20/250 TBS. All buffers contained 0.5 mM TCEP. Resulting ctSTIM1 samples were >90% pure and displayed the expected molecular weight (~52 kD) by SDS-PAGE. Dimer concentration was typically 25–50 µM, measured by absorption photometry (NanoDrop 2000, ThermoFisher Scientific).

In preparing intra-subunit smFRET samples, we took several measures to minimize the proportion of fluorescent homodimers (FF, where F is a double-labeled subunit) while optimizing the yield of heterodimers (NF, where N is an unlabeled subunit). To form heterodimers, double-cysteine ctSTIM1 and cysteine-free ctSTIM1 were denatured in buffer containing 8 M urea and then mixed in a 1:5 ratio. Refolding was allowed to take place during slow dialysis as described above, to obtain expected ratios of 1/36 FF, 10/36 NF, and 25/36 NN (unlabeled homodimers). The small FF fraction was excluded during post hoc analysis of the single-molecule movies by selecting only molecules with two fluorophores (see 'smFRET data analysis' below). In a subset of samples, we added an AviTag to the C-terminus of unlabeled ctSTIM1. After recombination with double-labeled ctSTIM1, the biotinylated AviTag was then used to selectively attach only NF and NN dimers directly to the surface of the imaging chamber (instead of encapsulating ctSTIM1 in liposomes), in principle eliminating any FF dimers.

Protein labeling was performed as described previously (*Choi et al., 2010*) using Alexa Fluor 555 as donor fluorophore and Alexa Fluor 647 as acceptor fluorophore (Invitrogen Life Technologies). For symmetric inter-subunit smFRET and for intra-subunit smFRET, donor and acceptor fluorophores were mixed in a 1:1 ratio before adding to the protein. Labeling efficiency assessed by absorption photometry was typically ~70 % but varied for different locations; cysteine mutants with labeling efficiency below 50 % were not used for smFRET experiments.

For asymmetric inter-subunit smFRET, two different ctSTIM1 cysteine mutants were labeled separately with donor or acceptor fluorophores and recombined afterward into ctSTIM1 heterodimers. For recombination, labeled ctSTIM1 homodimers were denatured with 8 M urea, mixed in a 1:1 ratio, and refolded during slow dialysis as described above, resulting in a mixture containing 50 % ctSTIM1 heterodimers. ctSTIM1 heterodimers with a single donor and a single acceptor fluorophore were individually selected during post hoc analysis of the smFRET experiments.

## Sample preparation

Proteins were either encapsulated in liposomes or attached directly to the coverslip substrate. Liposome encapsulation is often preferred to minimize surface effects, although several comparisons showed no significant difference between the FRET profiles of encapsulated and directly attached ctSTIM1 (417:417′ in *Figure 1—figure supplement 1*; 239:239′, 274:274′ in *Figure 1—figure supplement 2*; 242:431 in *Figure 1—figure supplement 3*). Except for the intramolecular measurements in *Figures 2 and 4*, all data were obtained using encapsulated protein. For liposome encapsulation, biotinylated lipids (18:1 Biotinyl Cap PE, Avanti Polar Lipids) were mixed with egg-PC lipids (Avanti Polar Lipids) in a 1:100 ratio and suspended in 20/150 TBS, and labeled protein was added to give final concentrations of ~2 mg/ml egg-PC and ~300 nM ctSTIM1. This mixture was passed back and forth (Extruder Set, Avanti Polar Lipids) through a liposome extrusion membrane (PC Membranes 0.1 µm, Avanti Polar Lipids). Liposomes were separated from free protein on a Sepharose CL-4B column (Sigma Aldrich). The conditions resulted in on average <<1 ctSTIM1 dimer per liposome.

Flow cells for single-molecule fluorescence imaging were prepared according to established protocols (*Roy et al., 2008*). Briefly, strips of double-sided tape were applied to a quartz microscope slide (Finkenbeiner) to form channel walls, with holes at the ends of each channel to create entry and exit points for buffer solutions. A microscope coverslip (Erie Scientific) was pressed on the tape strips and edges of the channels were sealed with epoxy glue. For liposome experiments channels were coated with 1 mg/ml BSA-biotin (Sigma Aldrich). For direct surface attachment microscope slides and coverslips were coated with a 100:1 PEG/PEG-biotin mixture (Laysan Bio) prior to flow cell construction. In both cases, channels were flushed with 0.2 mg/ml neutravidin (Thermo Fisher CWA), then rinsed with 20/150 TBS (150 mM NaCl, 20 mM Tris, pH 7.4 with HCl) before loading liposomes or biotinylated ctSTIM1 (~100 pM). Immediately prior to imaging, channels were filled with 20/150 TBS containing 100 µM cyclooctatetraene (Sigma Aldrich) and an oxygen scavenging system consisting of 1 % D-glucose, 1 mg/ml glucose oxidase (Sigma Aldrich), and 0.04 mg/ml catalase (Sigma Aldrich).

## TIRF microscopy and smFRET measurements

Single-molecule FRET recordings were performed using a home-built through-the-objective total internal reflection fluorescence (TIRF) imaging system based on a Zeiss Axiovert S100 TV microscope equipped with a Fluar 100 × 1.45 NA oil-immersion objective (Zeiss). Alexa Fluor 555 (donor) and Alexa Fluor 647 (acceptor) fluorophores were excited by 532 and 637 nm wavelength lasers (OBIS 532 nm LS 150 mW, Coherent and OBIS 637 nm LX 140 mW, Coherent). Donor and acceptor emission were separated by a 652 nm dichroic beamsplitter (Semrock) and passed through 580/60 nm and 731/137 nm bandpass filters (Semrock) mounted in an OptoSplit-II beamsplitter (Cairn Research), projecting the images side-by-side onto an EM-CCD camera (iXon DU897E, Andor). Hardware and image acquisition were controlled by BeanShell scripts in µManager (*Edelstein et al., 2010*), and image sequences were stored as stacked TIFF files.

Prior to image acquisition, the channel surface was scanned manually using low-intensity donor excitation to identify a suitable location for smFRET imaging, where the density of fluorescent spots appeared homogeneous and without contamination or aggregations. Donor and acceptor emission were then recorded in response to donor excitation, typically for 60 s, immediately followed by 1 s of acceptor excitation to directly identify acceptor fluorophores. Images were acquired in frame-transfer mode with a 100 ms integration time, and laser intensity was set to obtain a trade-off between good signal-to-noise ratio and sufficient active time for both fluorophores (ideally tens of seconds), while inducing fluorophore bleaching within the recording time window. Typically, 10–20 movies were recorded in each of four channels per microscope slide. Measurements were performed at room temperature.

## smFRET data analysis

Fluorescence movies were analyzed using custom Python scripts. Donor and acceptor images were aligned using pre-recorded registration images, and fluorescing molecules were identified in the donor and acceptor channels by detecting local signal maxima within a five-pixel-diameter neighborhood. Donor signals that could not be matched to any acceptor signal were discarded. Pixel values for each molecule were summed, and signals from sequential movie frames were stacked to construct the fluorescence time series for each molecule. Raw donor and acceptor signals were background-corrected

using the median fluorescence values within a circular 35-pixel-diameter region near each molecule in each frame. Leakage of donor signal into the acceptor channel in our system (approximately 7 % of the measured donor signal) was subtracted from the background-corrected acceptor signal.

γ correction was used to correct for differences in photon emission and detection probabilities of donor and acceptor fluorophores. The value of γ was empirically determined for each individual molecule (*McCann et al., 2010*), and the correction was applied by multiplying the donor fluorescence signal by γ before calculating the smFRET ratio. The average γ value from all 12,252 molecules was 1.1.

Traces were selected for final analysis if all of the following criteria were met: (1) the signal-to-noise ratio was ≥5; (2) the acceptor bleached in a single step before the donor bleached; (3) the γ factor was between 0.5 and 2.5; (4) spontaneous fluctuations of donor and acceptor fluorescence were negatively correlated; and (5) if a donor bleach event was recorded, it occurred in a single step.

The FRET ratio E was calculated at each time point prior to bleaching as $E = I_A/(I_A + \gamma I_D)$, where $I_A$ and $I_D$ are the respective acceptor and donor fluorescence values. A FRET histogram was constructed for each trace by distributing FRET amplitudes into 30 bins in a [–0.25, 1.25] range. Histograms were normalized by dividing each bin by the total number of FRET points in the trace. An ensemble histogram for multiple molecules was constructed by summing the normalized histograms of the individual traces and dividing each bin by the total number of molecules.

The part of the fluorescence traces where both donor and acceptor were active was parameterized by fitting a Hidden Markov model (HMM), using routines from the SMART software package (*Greenfeld et al., 2012*). To determine the optimal kinetic model, the Bayesian information criterion was used to select between models with different numbers of states. To compare FRET levels among different molecules in an ensemble, each HMM state was assigned to a cluster based on its mean FRET value. Clustering was performed based on all observed states in the ensemble, using a k-means algorithm with a predetermined number of target clusters. The number of clusters was determined ad hoc depending on the shape of the FRET histogram. For each cluster, the peak of its histogram was reported as its representative FRET level. The predominant FRET cluster was the cluster containing the most FRET measurement points.

Periods between cluster transitions were used to construct dwell time histograms. The first and last period of each trace were discarded, so that traces with fewer than two transitions were not included in the dwell time analysis. For transition density plots each transition was described by a coordinate given by the FRET level just before the transition (horizontal axis) and the level after the transition (vertical axis). The local transition density was then calculated on a 50 × 50 grid ranging from FRET –0.25–1.25, by convolution with a Gaussian kernel with two-pixel standard deviation. To prevent individual molecules with many transitions from dominating the plot, each molecule contributed only one data point for each kind of transition. In this way, the transition plot reports the number of molecules in which each kind of transition appeared.

For each protein sample, single-molecule recordings from multiple experiments were pooled to construct smFRET amplitude histograms. The total number of molecules sampled for each histogram (see *Figure 1—figure supplements 1–3*) was determined such that additional molecules did not alter the position of the predominant peak in the histogram. Key smFRET results were replicated by multiple authors (biological replications with newly purified protein samples). Molecules that did not produce a detectable level of smFRET were excluded, as they could not be properly interpreted.

## Molecular modeling and simulation

Molecular dynamics simulations and model optimization procedures were adapted from published procedures (*Choi et al., 2010*), and performed using Crystallography and NMR System (CNS) (*Brunger et al., 1998*; *Brunger, 2007*). Structures were visualized and manipulated using PyMOL (Version 1.8, Schrödinger, LLC.) and ProDy (*Bakan et al., 2011*). To predict distances between fluorophores, we performed molecular dynamics simulations using atomic models of donor and acceptor fluorophores. Because structures for Alexa Fluor 555 and Alexa Fluor 647 are not available, we used the structures of the spectrally similar Cy3 and Cy5. Residues of interest on the CAD structure were mutated to cysteine and fluorophore models were covalently attached by CNS. While all protein atoms were held fixed in space, a simulated-annealing protocol sampled possible fluorophore conformations until a local energy minimum was reached. This procedure was repeated 100 times to obtain a distribution

of fluorophore orientations, and the average location of the fluorophore center (taken as the location of the CAO atom) was computed from the resulting coordinates. When attached to sites in the CAD crystal structure the overall mean dye center protruded from the Cα atom of the attachment residue by 0.99 ± 0.08 nm for Cy3 and 1.10 ± 0.08 nm for Cy5. Since labeling was stochastic, the donor or acceptor could be on either side in a pair of label sites. To account for this, we averaged Cy3 and Cy5 center coordinates for each residue into a single 'effective' center location. Fluorophores were represented in the simulations by a pseudo-atom positioned at this effective center location, and inter-fluorophore distances were then calculated as the distance between two pseudo-atoms.

According to Förster theory the distance between donor and acceptor fluorophores ($R$) can be derived from experimentally determined smFRET efficiency ($E$) as $R = R_0 (1/E - 1)^{1/6}$, where $R_0$ is a proportionality factor reflecting the inter-fluorophore distance at which $E = 0.5$ (***Stryer and Haugland, 1967***). For Alexa Fluor 555 and Alexa Fluor 647 in water, the theoretical value of $R_0$ is 5.1 nm, assuming fast isotropic re-orientation of the fluorophores. We used this value for calculating smFRET-derived distances.

For structural modeling of the CAD apex (***Figure 2E***), we first created a symmetric version of the CAD crystal structure, as the original is slightly asymmetric in the apical region. We copied one subunit and aligned the proximal CC2 (aa 345–378) and CC3 (aa 408–436) domains to the adjacent subunit in the dimer, resulting in a symmetric pair. We then used CNS to allow the apex to relax to a new conformation, guided by smFRET-derived distance constraints (inter-subunit constraints at aa 388, 389, 399, 400, and 401, and intra-subunit constraint 431:389 on both subunits). Helices were treated as rigid bodies, while unstructured regions were treated as flexible chains with variable torsion angles. The distal CC2 helix (aa 379–391) was allowed to rotate around residue G379. During the relaxation of the apex, the proximal CC2 and CC3 domains were held fixed in space.

To reconstruct the arrangement of CC1 helices around CAD, we used the symmetric CAD structure with smFRET-optimized apical region and modeled CC1 as a chain of dimensionless nodes using a custom Python script. Each node on the chain represented the center of a CC1 residue. CC1 chains were generated starting from aa 344 at the CAD N-terminus, by progressively connecting new nodes until the final residue aa 233 was reached. In unstructured regions, nodes were spaced every 0.38 nm and were generated at random angles ≥90°. Alpha-helical regions (CC1α1–3, comprising residues aa 246–271, 275–305, and 310–337, respectively) were modeled as a straight chain of nodes at 0.15 nm intervals. To create a symmetric pair of CC1 chains, nodes were generated for one subunit, and then copied and mirrored in CAD's symmetry planes to create the chain for the other subunit. Each node was subjected to two criteria: (1) it cannot overlap with any residues in CC1 or CAD (volume exclusion in a 0.25 nm-radius sphere), and (2) it must satisfy smFRET-derived distance ($D_{FRET}$) constraints.

$D_{FRET}$ values could not be used directly for CC1 modeling because fluorophores were not explicitly simulated on CC1 residues. Instead, the 1 nm fluorophore linker length was accounted for by accepting distances within a range around $D_{FRET}$, determined as follows (***Supplementary file 1***). For CC1:CAD pairs, the modeled distance was calculated between the center of the CC1 node and the center of the simulated fluorophore on CAD. The applied distance range had a lower bound of $D_{FRET} -$ 1 nm, allowing for the possibility that the CC1 fluorophore was directed away from the corresponding fluorophore on CAD. For CC1:CC1′ pairs, the distance was calculated between the centers of the two CC1 nodes. The applied distance range had a lower bound of $D_{FRET} - 2$ nm to allow for the possibility that the two CC1 fluorophores were directed away from each other. In all cases, the upper distance bound was initially set to $D_{FRET}$ to exclude solutions where fluorophores were directed toward each other and the interior of the protein. Finally, lower and upper distance bounds were increased to reflect an uncertainty of ±0.05 (the bin width of the smFRET histograms) in the measurement of the smFRET peak.

If a model node fell within the distance range without overlapping other residues, it was retained; otherwise, it was rejected and a new node generated. If no acceptable node was found, a new search was started for the preceding node. If, after multiple trials, solutions could not be obtained for a particular smFRET measurement, the upper bound of the acceptable distance range was increased in 1 nm increments, until solutions could be obtained. Such adjustment was performed for 7 out of 36 measurements, 4 of which had very low FRET ( < 0.2). This process continued until the final N-terminal residue of CC1 (aa 233) was reached. Solutions obtained in this way together defined a solution space

within which CC1 obeyed the imposed constraints. We used the average of 50 solutions as a model for presentation (*Figure 5* and *Figure 5—figure supplement 1*).

To obtain a more detailed picture for the molecular interface of the CC1α1-CAD complex, we aligned the crystal structure of the CC1α1 helical segment (aa 246–271) to the smFRET-derived CC1 model, directing the hydrophobic sidechains to point toward CAD. Residue A419 in the CAD crystal structure was mutated back to the original valine using PyMOL. The CC1α1 helix was then docked onto CC3 using the flexpepdock server of Rosetta (*London et al., 2011*), which performed a molecular dynamics simulation with a fully flexible backbone. An initial docking was performed with distance constraints (2.5 ± 2.5 Å) between CG atoms of residue pairs L248:L416, L251:L416, L258:L423, and L261:L427, which were directly apposed in the smFRET-derived model. This was followed by an unconstrained simulation to obtain the final model (*Figure 5E*).

## Chemical crosslinking and mass spectrometry

For $BS^3$ crosslinking of ctSTIM1, the buffer was changed from Tris- to HEPES-based (150 mM NaCl, 20 mM HEPES, pH 8.0 with NaOH) during the ion exchange procedure. ctSTIM1 samples at 10 μM (~1 mg/ml) were allowed to incubate with 0, 20 and 50 μM $BS^3$ ($BS^3$-d0 or $BS^3$-d4, Sigma Aldrich) for 30 min at room temperature. The reaction was then quenched for 15 min at room temperature with 20 mM Tris.

For crosslinking with EDC, the buffer was changed from Tris- to MES-based (250 mM NaCl, 100 mM MES, pH 6.0 with HCl) during the ion exchange procedure. To increase crosslinking efficiency, EDC (Sigma Aldrich) was used in combination with sulfo-NHS (Sigma Aldrich) to convert carboxyl groups into stable amine-reactive sulfo-NHS esters. ctSTIM1 samples at 100 μM (~10 mg/ml) were allowed to incubate with 0/0, 20/62, 50/156, and 100/312 mM EDC/sulfo-NHS for 15 min at room temperature. Two volumes of PBS (250 mM NaCl, 100 mM $Na_2HPO_4$, pH 8.5 with NaOH) with 25 mM BME (pH 8.5) were then slowly added to quench the EDC and increase the pH for the subsequent amine reaction, incubated 2 hr at room temperature and quenched with 40 mM Tris.

Crosslinked ctSTIM1 samples were run on SDS-PAGE gels, and monomer and dimer bands were cut and analyzed separately by mass spectrometry (*Figure 5—figure supplement 2*). For comparative $BS^3$ crosslinking experiments, $BS^3$-d0- and $BS^3$-d4-treated samples were first mixed 1:1 before electrophoresis. Gel bands at 20 and 50 μM $BS^3$ were combined for analysis (*Figure 5—figure supplement 2*). In *Figure 7—figure supplement 2*, the T307C sample was first cysteine-crosslinked with CuP before treatment with $BS^3$-d4. After $BS^3$-d4 treatment the cysteine crosslink was reversed by treatment with DTT before mass spectrometry analysis. For EDC experiments, the monomer band at 10 mM EDC and the combined dimer bands at 2, 5, and 10 mM were used (*Figure 5—figure supplement 3*).

Mass spectrometry analysis of cross-linked samples was performed as described previously (*Komolov et al., 2017*). Briefly, gel bands were diced into 1 × 1 mm squares, reduced in 5 mM DTT at 55 °C for 30 min, and then alkylated with 10 mM acrylamide for 30 min at room temperature to cap free cysteines. Following alkylation of free cysteines and washing of gel pieces, proteolytic digestion was completed using trypsin/LysC protease (Promega) in the presence of ProteaseMax (Promega) overnight at 37 °C. Peptides were then extracted and dried under SpeedVac prior to LC/MS analysis.

Mass spectrometry experiments were performed using an Orbitrap Fusion Tribrid mass spectrometer (Thermo Scientific, San Jose, CA) with an Acquity M-Class UPLC system (Waters Corporation, Milford, MA) for reverse phase separations. The column was an in-house pulled-and-packed fused silica column with an I.D. of 100 microns pulled to a nanospray emitter. Packing material for the column was C18 reprosil Pur 1.8 micron stationary phase (Dr. Maisch) with a total length of ~25 cm. The UPLC system was set to a flow rate of 300 nL/min, where mobile phase A was 0.2 % formic acid in water and mobile phase B was 0.2 % formic acid in acetonitrile. Gel-extracted peptides were directly injected onto the column with a gradient of 2–45% mobile phase B, followed by a high-B wash over a total 90 min. The mass spectrometer was operated in a data-dependent mode using HCD/ETD decision-tree fragmentation in the orbitrap (HCD) or ion trap (ETD) for MS/MS spectral generation.

The collected mass spectra were analyzed using Byonic v2.12.0 or v2.14.27 (Protein Metrics) as the search engine for peptide ID and protein inference. The precursor and fragment ion tolerance were both set to 12 ppm and 0.4 Da for HCD and ETD, respectively. The search assumed tryptic digestion and allowed for up to two missed cleavages. Potential crosslinked peptides were identified using Byonic's xlink feature, allowing for linker mass where appropriate. Data were validated using

the standard reverse-decoy technique at a 1 % false discovery rate and inspected using Byologic for crosslink verification and assessment.

## Cysteine crosslinking

Symmetric inter-subunit ctSTIM1 dimers (5 µM) were incubated with 300 µM $CuSO_4$ and 900 µM o-phenanthroline for 15 min at room temperature, after which the reaction was quenched with 2 x LDS buffer containing 50 mM EDTA. After samples were run on an SDS-PAGE gel and stained with Coomassie Blue, gels were destained and scanned with a LI-COR Odyssey infrared imaging system. Crosslinking efficiency was quantified by the integrated intensity of the dimer band as a fraction of the summed intensities of the monomer and dimer bands.

For crosslinking flSTIM1 in situ, HEK293 cells were transfected with mCh-STIM1-C437S constructs with selected residues replaced by cysteine. 48 hr after transfection, cells were exposed to 0.2 mM diamide for 10 min in either 2 Ca Ringer's (full $Ca^{2+}$ stores) or in 0 $Ca^{2+}$ + 1 mM EGTA +20 µM cyclo-piazonic acid (depleted $Ca^{2+}$ stores; CPA, Sigma). Cells were then lysed in RIPA containing 20 mM NEM, protease inhibitor cocktail (Cell Signaling Technology) and EDTA. Samples were run on SDS-PAGE and analyzed by western blot using mouse anti-mCherry primary antibody (1:2000, Takara Bio) and secondary antibody (LI-COR) on a LI-COR Odyssey imaging system. 50 mM DTT was added to duplicate samples to test for disulfide crosslinks. Sites of interest were replicated 2–3 times (biological replicates with newly transfected cells).

## Calcium imaging

For experiments with STIM1 fragments (*Figure 6—figure supplement 1*), HEK293 cells were transfected with mCherry-labeled STIM1 cytosolic fragments (0.25 µg + 0.25 µg empty pcDNA3 vector) and Orai1-GFP (0.5 µg) in a 35 mm dish using Lipofectamine 2000 (Life Technologies). Transfected cells were grown overnight in DMEM supplemented with Glutamax, 10 % FBS, pen/strep, and 20 µM $LaCl_3$ to minimize the toxic effects of constitutive $Ca^{2+}$ influx. Cells were loaded with 1 µg/ml fura-2/AM for 30 min at room temperature, rinsed, and plated on poly-D-lysine-coated cover-slips for imaging. For experiments with flSTIM1 (*Figure 7*), HEK293 cells were co-transfected with flSTIM1 cysteine mutants and Orai1-GFP. 24–48 hr after transfection, cells were loaded with 1 µM fura-2/AM for 30 min at room temperature, rinsed, and plated on poly-D-lysine-treated coverslip chambers.

Fura-2 imaging was conducted using a Zeiss 200 M inverted microscope with a Fluar 40 X NA 1.3 objective; cells were excited alternately at 350 and 380 nm (Polychrome II, TILL Photonics), and emission at 534 ± 30 nm (Semrock) or >480 nm (Chroma) was collected with a Flash4.0 sCMOS camera (Hamamatsu Corp.) with 2 × 2 binning. Images were background corrected before calculating the 350/380 ratio for each cell. Standard bath solution contained (in mM): 155 NaCl, 4.5 KCl, 2 $CaCl_2$, 1 $MgCl_2$, 10 D-glucose and 5 Na-HEPES (pH 7.4). $Ca^{2+}$-free solution was prepared by replacing $CaCl_2$ with 2 mM $MgCl_2$ and 1 mM EGTA. For cysteine crosslinking experiments, cells were treated with 1 µM ionomycin +0.25 mM diamide in $Ca^{2+}$-free Ringer's followed by $Ca^{2+}$-free Ringer's + 0.25 mM diamide +0.1 % BSA to wash out ionomycin prior to readding standard (2 mM $Ca^{2+}$) Ringer's + 0.1 % BSA. Calcium imaging was performed on a sufficient number of cells to obtain an error of the mean that allowed a clear distinction among responses to different conditions. Cells with an abnormally high or unstable baseline calcium level were excluded.

## Acknowledgements

The authors thank members of the Lewis lab for helpful discussions during the course of this project. This work was supported by NIH grant R37GM45374, the Mathers Charitable Foundation, and a Stanford Medicine Discovery Innovation Award (RSL), and by NIH grant R37MH63105 (ATB). SvD was supported by Rubicon postdoctoral fellowship 825.13.027 from the Dutch Research Council NWO, and postdoctoral fellowship 16POST30780015 from the American Heart Association. Mass spectrometry was performed in collaboration with the Vincent Coates Foundation Mass Spectrometry Laboratory, Stanford University Mass Spectrometry. We thank R Leib and K Singhal for mass spectrometry analysis of crosslinked STIM1. The work was supported in part by NIH P30 CA124435 utilizing the Stanford Cancer Institute Proteomics/Mass Spectrometry Shared Resource.

# Additional information

### Competing interests

Axel T Brunger, Richard S Lewis: Reviewing editor, eLife. The other authors declare that no competing interests exist.

### Funding

| Funder | Grant reference number | Author |
| --- | --- | --- |
| National Institutes of Health | R37GM45374 | Richard S Lewis |
| Mathers Foundation | | Richard S Lewis |
| Stanford University | Discovery Innovation Award | Richard S Lewis |
| National Institutes of Health | R37MH63105 | Axel T Brunger |
| Nederlandse Organisatie voor Wetenschappelijk Onderzoek | 825.13.027 | Stijn van Dorp |
| American Heart Association | Postdoctoral fellowship 16POST30780015 | Stijn van Dorp |

The funders had no role in study design, data collection and interpretation, or the decision to submit the work for publication.

### Author contributions

Stijn van Dorp, Conceptualization, Data curation, Formal analysis, Funding acquisition, Investigation, Methodology, Software, Validation, Visualization, Writing – original draft, Writing – review and editing, Designed research; performed smFRET experiments and in vitro protein crosslinking assays; wrote data analysis software and performed data analysis and structural modeling; wrote and revised the manuscript; Ruoyi Qiu, Formal analysis, Validation, Writing – review and editing, Performed smFRET experiments; Ucheor B Choi, Formal analysis, Investigation, Methodology, Validation, Writing – review and editing, Performed smFRET experiments; Minnie M Wu, Formal analysis, Investigation, Visualization, Writing – review and editing, Performed in vivo cysteine crosslinking assays; Michelle Yen, Formal analysis, Investigation, Visualization, Writing – review and editing, Performed calcium imaging experiments with flSTIM1; Michael Kirmiz, Writing – review and editing, Formal analysis, Investigation, Performed calcium imaging with ctSTIM1 variants; Axel T Brunger, Funding acquisition, Resources, Supervision, Writing – review and editing; Richard S Lewis, Conceptualization, Funding acquisition, Methodology, Project administration, Resources, Supervision, Visualization, Writing – original draft, Writing – review and editing

### Author ORCIDs

Stijn van Dorp (iD) http://orcid.org/0000-0002-9940-7163
Ucheor B Choi (iD) http://orcid.org/0000-0003-1541-2967
Axel T Brunger (iD) http://orcid.org/0000-0001-5121-2036
Richard S Lewis (iD) http://orcid.org/0000-0002-6010-7403

### Decision letter and Author response

Decision letter https://doi.org/10.7554/eLife.66194.sa1
Author response https://doi.org/10.7554/eLife.66194.sa2

# Additional files

### Supplementary files

• Supplementary file 1. smFRET-derived distance measurements used to construct the CC1-CAD model. The table lists the complete set of amino acid pairs, their predominant smFRET efficiencies, the corresponding smFRET-derived distances, and the range of distance values used for generating

the models shown in *Figure 5A* and *Figure 5—figure supplement 1*. The 'FRET peak' and 'Distance' values correspond to the peaks of the smFRET histograms and the associated calculated distances shown in *Figure 1—figure supplements 1–3*. For sites with both liposome and avitag measurements, the liposome measurement was used to constrain the CC1:CAD model. The 'Range' values indicate the allowable distance bounds for each pair of modeled residues (see Materials and methods).

• Supplementary file 2. PyMOL file containing the stacked CC1-CAD models shown in *Figure 5—figure supplement 1*.

• Supplementary file 3. PyMOL file containing the wedged CC1-CAD models shown in *Figure 5—figure supplement 1*.

• Transparent reporting form

### Data availability

All data generated or analysed during this study are included in the manuscript and supporting files; Source Data files have been provided for Figure 1—figure supplement 1–3, Figure 2—figure supplement 1, Figure 3—figure supplement 1, Figure 4—figure supplement 1, Figure 5—figure supplement 1, Figure 5—figure supplement 2, Figure 5—figure supplement 3, Figure 7, Figure 7—figure supplement 1, and Figure 7—figure supplement 2. Custom code used to analyze smFRET data is available at https://github.com/vandorp/stim1_paper (copy archived at https://archive.softwareheritage.org/swh:1:rev:af5a8d03e6c788fd5ce4462ddddcdd4bac45950c).

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
