## [Editor Report]

This study uses complementary approaches to advance our mechanistic understanding of STIM1 activation, with elegant single molecule methods providing new details on STIM1 structure and dynamics. The data clarifies some of the controversy between domain packing in two differing X-ray and NMR structures and substantially contributes to a mechanistic and structural understanding of the STIM1 activation process.

---

## [Decision Letter]

**Decision letter after peer review:**

Thank you for submitting your article "Conformational dynamics of auto-inhibition in the ER calcium sensor STIM1" for consideration by *eLife*. Your article has been reviewed by 3 peer reviewers, including Marcel P Goldschen-Ohm as Reviewing Editor and Reviewer #1, and the evaluation has been overseen by Richard Aldrich as the Senior Editor. The following individual involved in review of your submission has agreed to reveal their identity: Yubin Zhou (Reviewer #2).

Essential revisions:

1) A discussion regarding the discrepancy between this study and a recent NMR structure of CC1 (PMID: 33106661) will strengthen the manuscript. The most notable difference lies in the positioning of CC1-a1 relative to the other two helices of CC1. Is this due to the missing of SOAR and other C-terminal sequences in the NMR structure?

2) What is the rationale for using 2 different methods for smFRET, either encapsuled or directly attached proteins? Were there any differences in smFRET for liposome encapsulated versus tethered ctSTIM1? Also, it is not clear which was used where throughout the paper. This needs to be clarified.

3) Regarding intra-subunit smFRET measurements, the 1:5 mixture means that 1 in 6 observations come from a dimer with double cysteine label sites in each subunit. How were these observations accounted for in the analysis?

4) Western Blots: There should be at least one negative control provided for the used ctSTIM1 and flSTIM1 lacking the cysteine introduced for crosslinking experiments in Figure 7 or Suppl. Figure 2 or 8.

5) Why are the bands for the fl STIM1 dimers in Figure 7A or Suppl. Figure 8b actually higher than expected from the monomer bands?

6) Please include at least a brief discussion of the rationale for selecting these aa in the CC1 domain tested for cysteine crosslinking?

7) In the discussion, line 367, the authors state "in the absence of CC1α3, the rigid CC1α2 helix would prevent CC1α1 from binding to CAD, which would then be free to interact with Orai1. Consistent with this explanation, deletion of CC1α2 in addition to CC1α3 allows the inhibitory clamp to function normally (Fahrner et al., 2014)." As it can be seen in the referenced manuscript Fahrner et al., STIM1 ΔCC1α3 is only slightly constitutively active and in addition activated in a store dependent manner. From the authors statement one would assume STIM1 ΔCC1α3 to be fully constitutively active. Additionally, this explanation would imply that deletion of CC1α2 would similarly prevent CC1α1 from binding to CAD as the deletion of CC1α3 does, but Fahrner et al., show that STIM1 ΔCC1α2 is inactive before store depletion.

8) In ctSTIM1 the CC1a2-CC1a3 linkers are close to each other (312:312') and disruption of their close association with R304W leads to activation (Figure 6). In contrast, based on cross linking at 307 in ctSTIM1 and flSTIM1 the authors suggest that the CC1a2-CC1a3 linkers are far apart in the resting conformation and move together upon activation (Figure 7). Clearly something more subtle than what is indicated by the cartoons is going on. Some discussion of this discrepancy between the cartoons and the data would be helpful.

---

## [Author Response]

Essential Revisions:1) A discussion regarding the discrepancy between this study and a recent NMR structure of CC1 (PMID: 33106661) will strengthen the manuscript. The most notable difference lies in the positioning of CC1-a1 relative to the other two helices of CC1. Is this due to the missing of SOAR and other C-terminal sequences in the NMR structure?

In our model, CC1α1 is aligned with CC3 and separate from CC1α2 and CC1α3, while Rathner et al., show CC1 in a 3-helix bundle. There are several possible explanations for these differences. In the Rathner et al., study, CC1 was studied in isolation, precluding interactions of CC1α1 with CC3 or more C-terminal parts of STIM1 as the reviewer notes. In addition, Rathner included 7 mM SDS to prevent oligomerization, which could also interfere with formation of CC1 dimers. Because CC1 was studied in isolation in the absence of its known interaction partners (CC1 and CAD), it is difficult to know with certainty whether the NMR structure exists in the context of the complete STIM1 protein, but an intriguing possibility is that it could represent a transient intermediate that mitigates the energetic cost of breaking helical interactions (CC1α1-CC3, CC1α2-CC1α3´) during STIM1 activation. We have added text describing the recent structure to the Introduction (lines 67-69) and a discussion of these points to the Discussion (lines 400-419).

2) What is the rationale for using 2 different methods for smFRET, either encapsuled or directly attached proteins? Were there any differences in smFRET for liposome encapsulated versus tethered ctSTIM1? Also, it is not clear which was used where throughout the paper. This needs to be clarified.

Both methods of attachment have been widely used in published smFRET studies. We used encapsulated proteins for most experiments and generally used direct attachment only for intra-subunit FRET experiments (where both dyes label a single subunit) to minimize the frequency of molecules with two labeled subunits. We did this by adding the avitag attachment site to the unlabeled subunit, so that the attached proteins are unlabeled homomers or labeled on only one subunit (see added text lines 523-535). For each smFRET sample the attachment method is indicated in Figure 1 —figure supplements 1-3. To confirm that the method of attachment did not influence the results, we compared the two methods for four samples (239:239´, 274:274´, 417:417´, and 242:431). As can be seen in Figure 1 —figure supplements 1-3, the major mode of the FRET histograms was not significantly affected by the attachment method. We now explain this in the Results (lines 106-109) and Methods (lines 551-557).

3) Regarding intra-subunit smFRET measurements, the 1:5 mixture means that 1 in 6 observations come from a dimer with double cysteine label sites in each subunit. How were these observations accounted for in the analysis?

Assuming random dimerization, the expected proportion of dimers with both subunits labeled is by our calculations 1/36; 10/36 will have 1 subunit labeled, and 25/36 will be unlabeled. Thus, 1 out of 11 fluorescent dimers have both subunits labeled; these were recognized by multiple bleaching steps for each dye label and were excluded from analysis. We selected only molecules that displayed a single photobleaching step for each dye. We have added an explanation of this to the Methods (lines 523-535).

4) Western Blots: There should be at least one negative control provided for the used ctSTIM1 and flSTIM1 lacking the cysteine introduced for crosslinking experiments in Figure 7 or Suppl. Figure 2 or 8.

We have added a representative control blot to Figure 7A showing the absence of detectable dimers in cells expressing WT flSTIM1 and treated with diamide. For ctSTIM1, the near absence of detectable crosslinking for K349C, N363C, and Q431C in Figure 2 —figure supplement 1A provides an upper limit to the level of background dimer formation.

5) Why are the bands for the fl STIM1 dimers in Figure 7A or Suppl. Figure 8b actually higher than expected from the monomer bands?

While monomers migrate as expected given their MW of ~100 kD, the STIM1 dimers migrate more slowly than expected. Given this result, we considered the possibility that STIM1 might form disulfides with a large protein in the cell, resulting in low mobility. This does not seem to be the case. Recently we have succeeded in purifying flSTIM1 protein, and find that purified flSTIM1 S339C forms dimers with the same anomalously low mobility after crosslinking by diamide in vitro (note that this protein lacks an added mCherry; thus the expected MW is 77 kD for the monomer and 154 kD for the dimer).

**Author response image 1. sa2fig1:** Dimers of flSTIM1 show anomalously low mobility in SDS-PAGE. Purified flSTIM1 was treated with 0.1 mM diamide for 10 min. In lane 2, 10 mM TCEP (reducing agent) was added to the crosslinked sample 10 min before running the gel. Coomassie blue SDS-PAGE with MW markers shown at right*.*

We do not know why the STIM1 dimers have such low mobility. One possibility is that the dimer may not be fully denatured under our conditions and thus may not bind SDS as completely as the monomer.

6) Please include at least a brief discussion of the rationale for selecting these aa in the CC1 domain tested for cysteine crosslinking?

We selected cysteine sites in the CC1 of flSTIM1 based on several sites in ctSTIM1 that formed crosslinks (aa 266, 274, 307, 309, and 339). In addition, given evidence that the ER-proximal CC1α1 domain (aa 233-261) dimerizes upon flSTIM1 activation (Hirve et al., 2018), we performed a complementary cysteine scan of the distal CC1α1 region (aa 262-269) to search for additional dimerization sites. We now explain this rationale in the Results (lines 298-302).

7) In the discussion, line 367, the authors state "in the absence of CC1α3, the rigid CC1α2 helix would prevent CC1α1 from binding to CAD, which would then be free to interact with Orai1. Consistent with this explanation, deletion of CC1α2 in addition to CC1α3 allows the inhibitory clamp to function normally (Fahrner et al., 2014)." As it can be seen in the referenced manuscript Fahrner et al., STIM1 ΔCC1α3 is only slightly constitutively active and in addition activated in a store dependent manner. From the authors statement one would assume STIM1 ΔCC1α3 to be fully constitutively active. Additionally, this explanation would imply that deletion of CC1α2 would similarly prevent CC1α1 from binding to CAD as the deletion of CC1α3 does, but Fahrner et al., show that STIM1 ΔCC1α2 is inactive before store depletion.

We agree that our proposal about the spacing function of CC1α2 and CC1α3 was not entirely consistent with the results of Fahrner et al., While some degree of constitutive activity has been reported upon deletion of all or part of CC1α3 (Yang et al., 2012; Yu et al., 2013; Fahrner et al., 2014), the current measurements by Fahrner et al., clearly show that this is only partial, and deletion of CC1α2 alone did not affect basal or depletion-activated current. Accordingly, we have deleted this section.

8) In ctSTIM1 the CC1a2-CC1a3 linkers are close to each other (312:312') and disruption of their close association with R304W leads to activation (Figure 6). In contrast, based on cross linking at 307 in ctSTIM1 and flSTIM1 the authors suggest that the CC1a2-CC1a3 linkers are far apart in the resting conformation and move together upon activation (Figure 7). Clearly something more subtle than what is indicated by the cartoons is going on. Some discussion of this discrepancy between the cartoons and the data would be helpful.

We thank the reviewer for pointing this out. The confusion arises partly because cysteine crosslinking is a much more stringent test of proximity and orientation than FRET; i.e., cysteines that appear closely apposed based on FRET measurements may not be close enough to form disulfides. In addition, the cartoon in Figure 6 was a rough schematic, while that in Figure 7 was meant to mimic the structure we determined from smFRET (Figure 5). We have made the cartoon in Figure 7D more consistent with the others and added an explanation in lines 285-296.